# MMRR: Unsupervised Anomaly Detection through Multi-Level Masking and Restoration with Refinement

## Abstract

Recent state-of-the-art anomaly detection algorithms mainly adopt generative models or approaches based on deep one-class classification. These approaches have hyperparameters to balance the adversarial framework of the generative adversarial network and to determine the decision boundary of the classifier. Both methods show good performance, but their performance suffers from *hyperparameter sensitivity*. A new category of anomaly detection methods has been proposed that utilizes prior knowledge about abnormal data or pretrained features, but it is more generic *not to use such side information*. In this study, we propose *"Multi-Level Masking and Restoration with Refinement (**MMRR**)"*, an unsupervised-learning-based anomaly detection method based on a generative model that overcomes *hyperparameter sensitivity* and the need for side information. MMRR learns the salient features of normal data distributions through *restoration from restricted information via masking*, resulting in a better restoration of in-distribution data than out-of-distribution data. To overcome *hyperparameter sensitivity*, we ensemble restoration results from information restricted to *predefined multiple levels* instead of finding a single optimal restriction level, and propose a novel mask generation and refinement method to achieve hyperparameter robustness. Extensive experimental evaluation on common benchmarks (*i.e.,* MNIST, FMNIST, CIFAR10, MVTecAD) demonstrates the efficacy of the MMRR.

## 1 Introduction

Anomaly detection tackles the problem of detecting abnormal data with a distribution that is significantly different from normal data. It is an important task that enables machine learning algorithms to cope with unexpected distribution in real-word tasks such as self-driving or medical imaging. Anomaly detection problems are formulated assuming the unavailability of abnormal data during the training process; therefore, anomaly detection models cannot be trained for the original purpose of anomaly detection. With same context, *it is impossible to validate in advance whether a proposed model performs anomaly detection well during the training process*. This means that even if the anomaly detection ability of the model is significantly affected by the hyperparameter values, it is impossible to find the optimal hyperparameter value through validation. Therefore, a method with a robust anomaly detection performance is necessary that does not include hyperparameters that have a significant influence on anomaly detection performance.

Three deep-learning-based leading strategies have been proposed to solve anomaly detection. The first is using methods based on generative model which perform anomaly detection based on the efficiency of the proposed generative models in restoring data. Early generative-model-based methods failed in the anomaly detection task owing to the good generalization capability of the autoencoder [37, 2]. Furthermore, to solve this problem, many studies [39, 36, 1, 8, 30, 31] inspired by generative

adversarial networks (GANs) [15] have attempted to create autoencoders that can only restore normal data by limiting the generalization capability using an adversarial concept. The second leading strategy is using deep one class classification methods [20, 35, 16, 21], which try to find the smallest hypersphere surrounding only normal data in unsupervised manner. However, generative model-based methods that try to restore *only* normal data well and deep one class classification methods that try to find hypersphere surrounding *only* normal data have hyperparameters that have a significant impact on anomaly detection performance. We define *hyperparameter sensitivity* problem as having hyperparameters that significantly affect performance even in the nature of the anomaly detection field where abnormal data is not available.

The third leading strategy is using side-information-based methods, which utilize prior knowledge about the difference between normal data and abnormal data [17, 13, 12, 18, 3, 41, 45, 25, 47] or utilize features [4, 28, 38, 5, 33] obtained from pretrained networks. Side-information based methods have shown good performance on many benchmark datasets, but it is not common to know side informations that can help distinguish normal data from abnormal data. In addition, these methods suffer from massive performace degradation in a setting where used side information is not applied well.

In this paper, we propose a novel method, Multi-Level Masking and Restoration with Refinement (MMRR) that does not use side information, is based on a generative model, and avoids the *hyperparameter sensitivity* problem. The motivation behind our proposed method is that a network trained to restore normal data from limited information about normal data will learn the salient features of normal data. So that restoration from limited information succeeds for normal data and fails for abnormal data, which makes it possible to perform anomaly detection in terms of restoration. To this end, our method consists of the following two key components. First, masking, which is a process that uses a mask to obtain restricted information by restricting the remaining information except for the parts essential for restoration. Second, restoration, which is the process of restoring original data by using only the restricted information obtained through masking.

For MMRR to perform anomaly detection, it is necessary to find the optimal masking level that causes normal data to be restored successfully and restoration of abnormal data to fail: masking level is the degree to which the mask limits information. However, to avoid the *hyperparameter sensitivity* problem caused by the absence of abnormal data during training, we detected anomalies through ensembles at multiple masking levels rather than finding a single optimal masking level. Our novel mask generation method made it possible to ensemble at multiple masking levels by enabling the manual control of the masking level of the mask, which eliminated the need for adversarial loss. In addition, our mask generation method made the mask learnable such that the mask most helpful for restoration at the corresponding masking level was generated, which led to better anomaly detection performance.

However, our masking method compares the degree of restoration at the same masking level without considering the complexity of each data. Therefore, masking and restoration alone often restores simple abnormal data better compared with complex normal data, in which case anomaly detection fails. To solve this problem, we propose an additional refinement process that eliminates the difference in restoration caused by the difference in data complexity. Our contributions are as follows:

- **Hyperparameter robustness and Prior knowledge-free.** We resolve the hyperparameter sensitivity problem that previous studies had overlooked with the proposed Multi-Level Masking and Restoration. Also, we have empirically shown through experiments that Multi-Level Masking is robust to hyperparameters. Furthermore, our method doesn't need any prior knowledge.

- **Experiments on benchmark datasets.** Unlike existing studies, MMRR does not strive to obtain optimal anomaly detection by solving the hyperparameter sensitivity problem. Nevertheless, we introduced Refinement Network considering the intrinsic complexity of data, and obtained comparable performance to SOTA approaches.

## 2 Related Works

Classical methods proposed to solve anomaly detection include PCA [19], OC-SVM [40], SVDD [42], iForest [26], and KDE [7]. Most of them perform anomaly detection using hand-crafted simple functions. However, advancements in deep learning have made it easier to obtain richer and more

complex features of data, and thus many deep-learning-based anomaly detection studies have been conducted. The following three strategies are widely used deep-learning-based anomaly detection tequniques.

**Generative-model-based methods.** Methods based on generative model begin with the assumption that the generative model trained only with normal data will fail to restore abnormal data. However, Sakurada and Yairi [37] and An and Cho [2] have demonstrated that a, simple autoencoder and variational autoencoder sufficiently restore abnormal data, thereby leading to the failure of anomaly detection. Therefore, various autoencoders for anomaly detection have been proposed that perform certain tasks, such as denoising [36] and inpainting [48]. Also, there are studies that [8, 23] used backpropagation to measure the distance from the manifold of the data. Many generative-model-based methods are inspired by GAN and adversarial training. Some previous studies assumed networks that learned normal distribution through adversarial training would not be able to restore abnormal data or classify them as fake data [39, 1]. Some studies have highlighted that autoencoders have good generalization capabilities and tried to design autoencoders that have limited resotraion capability by limiting the latent space through adversarial loss [30, 31], or by prototyping the latent space [14, 20]. However, most generative-model-based methods suffer from the *hyperparameter sensitivity* problem because they have to find the optimal point that balances adversarial losses and other losses to obtain the best anomaly detection performance, which is impossible because of the absence of abnormal data.

**Deep one-class classification methods.** Since anomaly detection cannot use abnormal data for training, it is difficult to design a classifier that distinguishes between normal data and abnormal data. Ruff et al. [35] proposed a deep learning solution called SVDD [42] that seeks to find the smallest hypersphere surrounding normal data. They used various constraints to prevent representation collapse due to the absence of abnormal data during the training process. Hu et al. [21] proposed a constraint called holistic regularization to prevent representation collapse. Some studies have artificially generated abnormal data for training one-class classifiers. Goyal et al. [16] obtained, artificial abnormal data through adversarial search, and Pourreza et al. [32] utilized data generated from immature generator as abnormal data. Methods based on deep one class classification suffer from the *hyperparameter sensitivity* problem as there are variables that significantly influence the performance of anomaly detection, such as the radius variable in Goyal et al. [16].

**Side-information-based methods.** Self-supervised methods utilize prior knowledge of the differences between normal and abnormal data. For example, some studies [13, 18, 3, 41] focused on differences in terms of geometry. Golan and El-Yaniv [13] assumed that a network can learn the geometric features of normal data through a learning process that predicts the geometric transsformations applied to normal data. They expected the that a trained transform classifier will fail to predict abnormal data with different geometric characteristics compared with normal data. Based on this study, a method to restore transformed data [11] and methods that combined geometric concept with constructive learning [6] were proposed [3, 41]. Other self-supervised methods augment normal data to create synthetic abnormal data and use them to train networks that can detect locally defect areas [45, 25, 47]. However, as mentioned in Goyal et al. [16], these methods rely heavily on prior knowledge. Some studies have attempted to perform anomaly detection using features obtained from pre-trained networks using external data. [4, 28, 38, 5, 33, 9]

# 3 Multi-Level Masking and Restoration with Refinement

The overall framework of the proposed method is shown in Fig. 3. In this section, we provide a detailed description of our method called Multi-Level Masking and Restoration with Refinement (MMRR). We describe the *Multi-Level Masking* and *Restoration* procedures that restrict the information in a given input, and finally the *Refinement* that further improves the restored image.

## 3.1 Multi-Level Masking

Masking is a process that restricts embedding $e \in \mathbb{R}^d$, which is generated through embedding network($f_E : \mathbb{R}^d \to \mathbb{R}^d$) as $e = tanh(f_E(x))$, by using mask $m$. The masking process is

$$\tilde{e} = e \odot m + \epsilon \odot (1 - m), \tag{1}$$

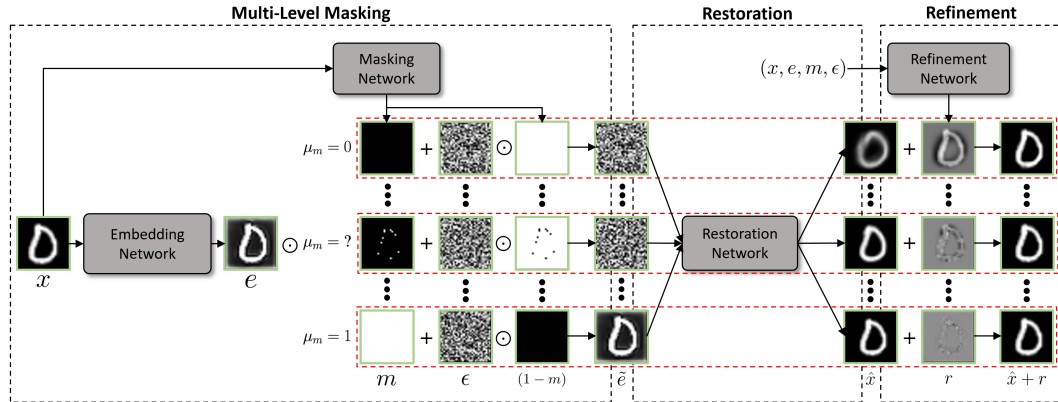

Figure 1: **Overall framework of MMRR.** Given data $x$, the embedding network $f_E$ generates embedding $e$. The embedding thus generated is limited through a mask $m$ with a masking level $\mu_m$ generated through the masking network $f_M$, and using only such restricted embedding, the restoration network $f_{\text{res}}$ performs the restoration of the original data $x$. Finally, the refinement network $f_{\text{ref}}$ complements the part not restored where restoration has inevitably failed due to the intrinsic complexity, which allows MMRR to perform anomaly detection only with the intended difference caused by masking and restoration.

where $\odot$ is the Hadamard product, and $\epsilon$ is noise sampled from uniform random noise $\epsilon \sim \mathcal{U}(-1, 1)^d$. The output of the masking process, $\tilde{e}$, is called restricted embedding.

We masked $e$ instead of directly masking $x$ because the training process using only normal data will make $f_E$ generate $e$, which helps in restoration. Thus, using $e$ will enable our proposed masking and restoration method to have a better discrimination ability. Noise $\epsilon$ is used because, without $\epsilon$, irrespective how small $m$ is, trivial solution that can easily restore data is generated because $e$ is learnable. For the same reason, $tanh$ was used to create $e$ to prevent a trivial solution that makes restoration easier by making the value of $e$ significantly different from the noise value.

We can easily infer from Eq. 1 that if the value of $m$ become smaller, the portion of embedding $e$ in $\tilde{e}$ decreases and becomes noisy, and restoration becomes harder. For example, if all elements of $m$ are 0, $\tilde{e}$ will resemble uniform noise $\mathcal{U}(-1, 1)^d$, and restoration will be impossible. Therefore, we consider that the average value of $m$ can represent the difficulty of restoration from $\tilde{e}$ and define it as a masking level $\mu_m = \frac{1}{d} \sum_{i=1}^{d} m_i$, where $m_i$ refers to $i$-th element of $m$ and $\mu_m \in [0, 1]$.

The restricted embedding $\tilde{e} \in \mathbb{R}^d$ should meet two conditions for masking and restoration to detect anomalies: normal data should be successfullt restore and abnormal data should not be restored. To accomplish the goal, we need to find $m$ with a $\mu_m$ that can best differentiate normal and abnormal data in terms of restoration. However, we cannot find an optimal masking level $\mu_m$ that best distinguishes abnormal data from normal data. This is because abnormal data cannot be used in the training process owingto the nature of the field of anomaly detection.

Therefore, we decided to ensemble the ability to distinguish at multiple masking levels $\mu_m$, which are uniformly distributed between 0 and 1. For example, if we use $L$ levels of masking for the ensemble, $\mu_m \in \{0, \frac{1}{L-1}, \frac{2}{L-1}, ..., 1\}$ will be used.

Our novel mask generation method made it possible to manually adjust the $\mu_m$ of $m$ for multilevel ensemble. Furthermore, the novel mask generation made $m$ learnable such that it is generated in a direction that is most useful for restoration from the corresponding $\mu_m$, which improves the ability to distinguish between normal data and abnormal data.

**Mask generation method.** We propose a novel mask generation method that can generate a mask $m$ with masking level $\mu_m$ by $m = \sigma(f_M(x) + b)$, where $f_M : \mathbb{R}^d \to \mathbb{R}^d$ is the masking network. The goal of our mask generation method is to find the appropriate bias $b \in \mathbb{R}$ that makes the average value of the mask to a predefined $\mu_m$ as follows: $\frac{1}{d} \sum_{i=1}^{d} \sigma(f_M(x)_i + b) = \mu_m$, where $\mu_m$ is on interval $[0, 1]$ because $m \in [0, 1]^d$. As sigmoid $\sigma$ is a monotonically increasing function, we can use the root-finding method (in our case, the bisection method) to find bias $b$ that satisfies the condition,

which allows us to successfully generate mask $m$ with masking level $\mu_m$. While the root finding method is non-differential, the gradient to the output of $f_M$ was obtained under the assumption that the bias satisfying the condition was well found, which is as follows:

$$\frac{\partial \mathcal{L}}{\partial f_M(x)_i} = \sum_{\forall j} \frac{\partial \mathcal{L}}{\partial m_j} m_j(1 - m_j) \left( \delta(i,j) - \frac{m_i(1 - m_i)}{\sum_{\forall k} m_k(1 - m_k)} \right), \quad \delta(i,j) = \begin{cases} 1, & \text{if } i = j \\ 0, & \text{otherwise} \end{cases}.$$
(2)

## 3.2 Restoration

Restoration refers to the process in which restricted embedding $\tilde{e}$ is restored to original data $x$ via the restoration network($f_{\text{res}} \colon \mathbb{R}^d \to \mathbb{R}^d$), where the restoration output is $\hat{x} = tanh(f_{\text{res}}(\tilde{e}))$. Owing to the mask generation method, not only $f_{\text{res}}$ but also $f_E$ and $f_M$ can be trained with only the simple restoration loss, which is formulated as:

$$\mathcal{L}_{\text{res}} = \frac{1}{d} \sum_{i=1}^{d} (x_i - \hat{x}_i)^2$$
(3)

$f_{\text{res}}$, which is trained using a training dataset consisting of only normal data, learns how to restore normal data from $\tilde{e}$. In such a training process, $f_{\text{res}}$ will learn salient features for normal distribution $p^+$. The features for normal distribution $p^+$ obtained in this way will allow $f_{\text{res}}$ to restore normal data efficiently even when the masking level $\mu_m$ is small.

While $f_{\text{res}}$ has been able to successfully restore normal data as mentioned above, this way of restoration will fail for abnormal data. The reason is, $f_{\text{res}}$ has no choice but to generate an output that resembles normal data because $f_{\text{res}}$ will also apply learned features for $p^+$ even when restoration is performed from $\tilde{e}$ of abnormal data. This failure to restore abnormal data will allow the masking and restoration method to detect anomalies through the restoration loss.

## 3.3 Refinement

Our masking and restoration method resolves the *hyperparameter sensitivity* problem by ensembling the anomaly detection performance at multiple masking levels $\mu_m \in \{0, \frac{1}{L-1}, \frac{2}{L-1}, ..., 1\}$. However, comparing the degree of restoration at the same $\mu_m$ without considering the characteristics of the data causes another problem. This is because the degree of restoration is intrinsically different even if it is restored from the same $\mu_m$ because different $x$ have different complexities.

Let us assume that the restoration loss obtained from masking and restoration is composed of two losses. The first loss is caused by the inevitable restoration failure due to the intrinsic complexity of $x$, which is denoted as intrinsic loss. The second loss occurs when abnormal data are restored like normal data owing to masking and restoration, which is denoted as abnormality loss. We originally intended to perform anomaly detection based on this abnormality loss.

This problem occurs when the abnormal sample is relatively simple compared with the normal sample. In this case, the sum of the intrinsic loss and the abnormality loss of the abnormal sample can be smaller than the intrinsic loss of the normal sample, which leads to the anomaly detection failure of the masking and restoration method.

To address this problem, the refinement method aims to eliminate the intrinsic loss that inevitably occurs due to intrinsic complexity difference so that anomaly detection can be performed only with the abnormality loss caused by masking and restoration process. For this, the refinement network $f_{\text{ref}} : \mathbb{R}^d \times \mathbb{R}^d \times \mathbb{R}^d \times \mathbb{R}^d \to \mathbb{R}^d$ predicts $x - d$ that have not yet been restored at a particular $\mu_m$ as follows: $r = f_{\text{ref}}(x, e, m, \epsilon)$. $f_{\text{ref}}$ is trained with refinement loss formulated as:

$$\mathcal{L}_{\text{ref}} = \frac{1}{d} \sum_{i=1}^{d} (x_i - (\hat{x}_i + r_i))^2$$
(4)

## 3.4 Training and Evaluation

**Training.** Our method consists of a two-step training process. The first phase is a training process for masking and restoration. During this phase, the masking level $\mu_m$ is uniformly sampled, where

$\mu_m \sim \mathcal{U}(0,1)$. The embedding network $f_E$, masking network $f_M$, restoration network $f_{\text{res}}$ are trained only with restoration loss. Furthermore, we selected the model with the smallest restoration loss for the validation data. The second phase is a training process for refinement. To this end, networks that have been trained in the first phase are used with fixed weights. $\mu_m$ is sampled from $\mathcal{U}(0,1)$ as in first phase. The refinement network $f_{\text{ref}}$ is trained with only the refinement loss. Furthermore, we selected the model with the smallest refinement loss for the validation data.

**Evaluation.** For evaluation, we must first determine the number of $\mu_m$ required. If we decide to use $L$ masking levels, we must use $\{0, \frac{1}{L-1}, \frac{2}{L-1}, ..., 1\}$ masking levels that are distributed evenly at $1/(L-1)$ intervals for evaluation. Finally, we perform anomaly detection by summing the refinement loss at all masking levels for ensemble.

## 4 Experiment

### 4.1 Experimental Settings

To validate the proposed anomaly detection method, MMRR using multi-class datasets (MNIST [24], FMNIST [44], CIFAR10 [22]), which is not designated for anomaly detection, we used the one-vs-all strategy. The one-vs-all strategy selects one normal class $1 \leq c \leq C$ among $C$ different classes. For training, we only used the training set belonging to class $c$. For testing, the normality score was calculated for all the data in the test set, the extent to which normal data and abnormal data are distinguished in terms of the normality score was measured using the area under receiver operating curve (AUROC). This process was repeated for all classes $C$ to evaluate the anomaly detection model. On the other hand, in the case of the MVTecAD dataset, for each class $c$, the train dataset consisting of only normal data and the test dataset mixed with abnormal data are already prepared. Therefore, we trained using only the train data as a given material, and used the test dataset in the test process.

**Implementation details.** All proposed networks were implemented using the U-Net[34] based on the wide residual[46] blocks proposed for wide residual networks. We used group normalization for all blocks. For 32x32 datasets, we used four feature map resolutions(32x32 to 4x4). For 256x256 datasets, we used five feature map resolutions(256x256 to 16x16). We used two wide residual blocks that consisted of convolutions with 128 output channels for each feature map resolution. RAdam[27] was used as the optimizer with a learning rate of 0.0001. Batch size was set as 64 and 4 for the 32x32 and 256x256 datasets, respectively. The learning was decayed by a factor 0.5 if the validation loss did not decrease for 500 epochs. We split the normal training set into training and validation sets using a 95:5 ratio, and used the validation set to select the model with smallest validation loss.

### 4.2 Datasets and Results

**Baseline Methods.** For anomaly detection in multi-class datasets, we compared MMRR with classical approaches such as: OC-SVM [40], and KDE [29]; generative-model-based approaches such as: AnoGAN [39], OCGAN [30], $\gamma - VAE_g$ [10] and $CAVGA_u$ [43]; deep one-class classification approaches such as: DSVDD [35], and DROCC [16]. For anomaly detection on MVTecAD dataset, we compared our MMRR with vanilla autoencoder AE, AE with skip connectins AE+skip, variational autoencoder VAE, Ganomaly[1], MemAE [14], $CAVGA_u$, and DAAD [20].

- **MNIST** includes a training set of 60,000 examples, and a test set of 10,000 examples. The data are 28x28 handwritten digits(0-9). For simplicity they were resized to 32x32. It was used for training without any augmentations except resizing. Our MMRR model achieved averaged AUROC of 0.967, which is slightly lower compared to SOTA methods. The reason our model has slightly poor performance on the MNIST dataset is that the data have a very easy distribution, so reconstruction occurs well enough even at a very low masking level $\mu_m$. For example, in Figure 3, we can see that the digit 0 is restored well enough even if $\mu_m$ is 0.01. As such, if there is already a sample that can be restored well in the masking and restoration stage of very low $\mu_m$, it can be seen that refinement has a limit in solving this problem.

- **FMNIST** consists of a training set of 60,000 examples, and test set of 10,000 examples, full of 10 different types of fashion items. For simplicity they were resized to 32x32. It was used for training without any augmentations except resizing. MMRR greatly beats the existing SOTA performance of 0.885 AUROC of CAVGA.

- **CIFAR10** consists of 60000 32x32 color images in 10 classes, with 6000 images per class. There are 50000 training and 10000 test images. The dataset was used for training without any augmentations. As shown in Table 5, our method achieved an average AUROC performance of 0.737 on the CIFAR10 dataset, which is comparable to that of other SOTA methods: 0.742 for DROCC and 0.737 for CAVGA. Moreover, the performance obtained by our method is meaningful because it is obtained without experiencing *hyperparameter sensitivity* problem unlike other SOTA methods.

- **MVTecAD** is a dataset for benchmarking anomaly detection methods with a focus on industrial inspection. It contains over 5000 high-resolution images divided into 15 different object and texture categories. Each category comprises a set of defect-free training images and a test set of images with a variety of defects as well as images without defects. We resized all the data to 256x256. We performed two tasks on the MVTecAD dataset, image-level anomaly detection and pixel-level anomaly localization. Experimental results on MVTecAD dataset can be seen in Table 7. MMRR achieved average 0.865 AUROC for pixel-level anomaly detection and 0.844 AUROC for image-level anomaly detection, which is close to SOTA methods. We found that among the test defect-free data in the screw class, there were samples with a different distribution in terms of brightness compared to the train defect-free data. Therefore, we trained MMRR by applying brightness augmentation to the train data, and a result of 0.95 AUROC was obtained in the image-wise anomaly detection. However, we did not report the performance because we assumed that we do not know the distribution of the test data.

**Hyperparameter sensitivity.** As we mentioned earlier, most of the generative model based methods and deep-one class classification based methods have *hyperparameter sensitivity* problem. For example, DROCC [16] showed how sensitively the performance changes according to the radius value, which is a hyperparameter that they used to obtain negative samples. Anomaly detection performace of DROCC in CIFAR10 dataset fluctuates between 0.7-0.8 for airplane, 0.5-0.7 for deer, and 0.7-0.8 for trucks in terms of AUROC depending on the radius value. Therefore, they carefully searched for the radius value to obtain optimal anomaly detection performance. In addition to this, Akçay et al. [1] showed that the performance of their proposed model is sensitively changed according to the values of three hyperparameters that balance their losses in the CIFAR10 dataset. Also, Hou et al. [20] showed that the anomaly detection performance in MVTecAD dataset fluctuates between 0.716-0.821 based on the value of division rate($r_h$&$r_w$) that determines the size of the query.

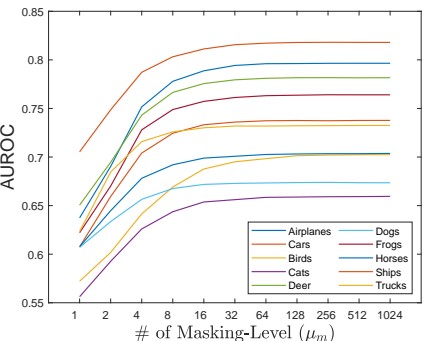

Figure 2: Illustration of AUROC with respects to the number of Masking-Level ($\mu_m$) used for MMRR on CIFAR10 dataset.

However, MMRR uses only one loss for each training phase. And we provide a clear criterion for model design: selecting the model with the lowest loss on validation data. Furthermore, we show how the performance of MMRR changes according to the only hyperparameter that significantly affects our performance in the Fig. 2. From Fig. 2, It can be seen that the performance of anomaly detection improves as the number of masking levels used for evaluation increases.

**Prior knowledge.** It has been shown in Goyal et al. [16] that the side-information based methods mentioned in Section 2 relies heavily on the prior knowledge they used. To prove this, they applied flips and small rotations of angle $\pm 30°$ to CIFAR10 data during training. As can be seen in Table 1 there was a large decline in the performance(0.86 to 0.691) of the Golan and El-Yaniv [13] that used prior knowledge. On the other hand, MMRR w/o refine showed rather good performance (0.676 to 0.682), and MMRR showed 0.037 lower performance (0.737 to 0.7).

|  | GEOM | MMRR w/o ref. | MMRR |
|---|---|---|---|
| w/o aug. | 0.86 | 0.676 | 0.737 |
| w/ aug. | 0.691 | 0.682 | 0.7 |

Table 1: Comparing AUROC against GEOM[13] on CIFAR10 dataset with training data augmentations (rotation $\pm 30°$ and flips).

### 4.3 Ablation Study

| MNIST | OC-SVM | KDE | AnoGAN | DSVDD | OC-GAN | CAVGA | MMRR w/o ref. | MMRR |
|---|---|---|---|---|---|---|---|---|
| 0 | 0.988 | 0.885 | 0.966 | 0.98 | 0.998 | 0.994 | 0.9857 | 0.9941 |
| 1 | 0.999 | 0.996 | 0.992 | 0.997 | 0.999 | 0.997 | 0.999 | 0.9982 |
| 2 | 0.902 | 0.71 | 0.85 | 0.917 | 0.942 | 0.989 | 0.8981 | 0.94 |
| 3 | 0.95 | 0.693 | 0.887 | 0.919 | 0.963 | 0.983 | 0.9246 | 0.955 |
| 4 | 0.955 | 0.844 | 0.894 | 0.949 | 0.975 | 0.977 | 0.9309 | 0.9352 |
| 5 | 0.968 | 0.776 | 0.883 | 0.885 | 0.98 | 0.968 | 0.9173 | 0.971 |
| 6 | 0.978 | 0.861 | 0.947 | 0.983 | 0.991 | 0.988 | 0.9765 | 0.989 |
| 7 | 0.965 | 0.884 | 0.935 | 0.946 | 0.981 | 0.986 | 0.9539 | 0.966 |
| 8 | 0.853 | 0.669 | 0.849 | 0.939 | 0.939 | 0.988 | 0.906 | 0.945 |
| 9 | 0.955 | 0.825 | 0.924 | 0.965 | 0.981 | 0.991 | 0.9511 | 0.98 |

Table 5: Image-level AUROC for one-vs-all anomaly detection on MNIST.

**Embedding.** To prove the effectiveness of using embedding $e$, we directly masked the data $x$. As can be seen from the table, we got an average AUROC of 0.6449 in the CIFAR10 dataset when $e$ was not used. And 0.6449 AUROC is far lower than 0.737 AUROC, which is the performance obtained when $e$ is used. Through these results, it can be seen that $f_E$ learned a salient features for normal data in the training process of generating $e$, which is most helpful for restoration even though it is restricted by masking. And the

|  | w/o ref. | w/ ref. |
|---|---|---|
| w/o emb. $e$ | 0.642 | 0.6449 |
| w/ emb. $e$ | 0.676 | 0.737 |

Table 2: AUROC performance of MMRR w/o and w/ embedding network on CIFAR10.

embedding $e$ generated from $f_E$ can be seen to have a positive effect on the anomaly detection performance by widening the restoration gap between normal data and abnormal data.

**Mask generation method.** We proved the effectiveness of our learnable mask by comparing it with other simple masks which are unable to learn. The first mask is a mask in which all elements have the same constant value $\mu_m$, and we will call it a constant mask. The second mask to be compared is a mask generated by bernoulli sampling with a probability of $\mu_m$. When we used the constant mask, we got an AUROC performance of 0.667, and when we used the bernoulli mask, we got 0.6478. These are lower performances when compared to 0.737 obtained by our mask generation method. From the experimental results, it can be seen that the use of a our multi-level mask that can learn to leave information which is most helpful for restoration at a specific masking level during the masking process also helps anomaly detection.

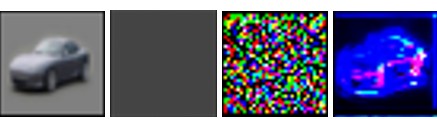

(a) From left to right, data, constant mask, bernoulli mask, our mask

|  | Constant | Bernoulli | Ours |
|---|---|---|---|
| w/o ref. | 0.619 | 0.612 | 0.676 |
| w/ ref. | 0.674 | 0.648 | 0.737 |

Table 3: AUROC according to mask generation method on CIFAR10

**Refinement.** As can be seen from the Table 4, there is a big difference between MMRR with refinement and MMRR without refinement. In the case of MNIST dataset, average auroc improved by 0.033 from 0.944 to 0.967. And for CIFAR10 dataset, average auroc improved by 0.067 from 0.68 to

|  | MNIST | FMNIST | CIFAR10 | MVTecAD |
|---|---|---|---|---|
| w/o ref. | 0.944 | 0.928 | 0.676 | 0.825 / 0.861 |
| w/ ref. | 0.967 | 0.93 | 0.737 | 0.840 / 0.865 |

Table 4: AUROC w/o and w/ refinement module on MNIST, FMNIST, CIFAR10, and MVTecAD. *Image-wise / Pixel-wise* AUROC performance was reported on MVTecAD.

0.747. Experimental results show another interesting phenomenon besides performance improvement. For example, data that has already had good anomaly detection performance in MMRR w/o refinement, such as data belonging to airplane, deer, ship classes, does not improve significantly when refinement is applied as can be seen in Table 5. However, the data that performed poorly in the MMRR w/o refinement, such as data belonging to automobile, truck, showed a remarkably large performance improvement. These results show that the intrinsic complexity difference between classes is well resolved through the refinement as intended. However, as MVTecAD dataset is the data proposed to detect local defect areas, the difference in intrinsic complexity between normal data and abnormal data is not large. Therefore, as can be seen from the Table 4, the performance improvement due to refinement was insignificant.

| CIFAR10 | OC-SVM | KDE | AnoGAN | DSVDD | OC-GAN | $\gamma$-VAE | CAVGA | DROCC | MMRR w/o ref | MMRR |
|---|---|---|---|---|---|---|---|---|---|---|
| Airplane | 0.63 | 0.658 | 0.671 | 0.617 | 0.757 | 0.702 | 0.653 | 0.817 | 0.7778 | 0.7965 |
| Automobile | 0.44 | 0.52 | 0.547 | 0.659 | 0.531 | 0.663 | 0.784 | 0.767 | 0.6065 | 0.7377 |
| Bird | 0.649 | 0.657 | 0.529 | 0.508 | 0.64 | 0.68 | 0.761 | 0.667 | 0.6926 | 0.7024 |
| Cat | 0.487 | 0.497 | 0.545 | 0.591 | 0.62 | 0.713 | 0.747 | 0.671 | 0.6076 | 0.6595 |
| Deer | 0.735 | 0.727 | 0.651 | 0.609 | 0.723 | 0.77 | 0.775 | 0.736 | 0.7638 | 0.7817 |
| Dog | 0.5 | 0.496 | 0.603 | 0.657 | 0.62 | 0.689 | 0.552 | 0.744 | 0.6143 | 0.6739 |
| Frog | 0.725 | 0.758 | 0.585 | 0.677 | 0.723 | 0.805 | 0.813 | 0.744 | 0.6966 | 0.7641 |
| Horse | 0.533 | 0.564 | 0.625 | 0.673 | 0.575 | 0.588 | 0.745 | 0.714 | 0.626 | 0.7037 |
| Ship | 0.649 | 0.68 | 0.758 | 0.759 | 0.82 | 0.813 | 0.801 | 0.800 | 0.7878 | 0.8181 |
| Truck | 0.508 | 0.54 | 0.665 | 0.731 | 0.554 | 0.744 | 0.741 | 0.762 | 0.6229 | 0.7325 |

Table 6: Image-level AUROC for one-vs-all anomaly detection on CIFAR10.

| | Method | carpet | grid | leather | tile | wood | bottle | cable | capsule | hazelnut | metalnut | pill | screw | toothbrush | transistor | zipper |
|---|---|---|---|---|---|---|---|---|---|---|---|---|---|---|---|---|
| Pixel-level | AE | 0.539 | 0.96 | 0.751 | 0.476 | 0.63 | 0.909 | 0.732 | 0.786 | 0.976 | 0.88 | 0.885 | 0.979 | 0.971 | 0.906 | 0.68 |
| Pixel-level | VAE | 0.58 | 0.888 | 0.834 | 0.465 | 0.695 | 0.902 | 0.828 | 0.862 | 0.977 | 0.881 | 0.888 | 0.958 | 0.971 | 0.894 | 0.814 |
| Pixel-level | $\gamma - VAE_g$ | 0.727 | 0.979 | 0.897 | 0.581 | 0.809 | 0.931 | 0.88 | 0.917 | 0.988 | 0.914 | 0.935 | 0.972 | 0.983 | 0.931 | 0.871 |
| Pixel-level | MMRR w/o ref. | 0.6733 | 0.8529 | 0.8599 | 0.7851 | 0.7911 | 0.8878 | 0.9117 | 0.898 | 0.8555 | 0.8648 | 0.9335 | 0.9074 | 0.9506 | 0.8865 | 0.8526 |
| Pixel-level | MMRR | 0.6561 | 0.8477 | 0.8405 | 0.7916 | 0.7858 | 0.889 | 0.8841 | 0.9179 | 0.9414 | 0.8197 | 0.9209 | 0.8924 | 0.9486 | 0.9038 | 0.8779 |
| Image-level | Ganomaly | 0.699 | 0.708 | 0.842 | 0.794 | 0.834 | 0.892 | 0.757 | 0.732 | 0.785 | 0.7 | 0.743 | 0.746 | 0.653 | 0.792 | 0.745 |
| Image-level | AE | 0.411 | 0.841 | 0.615 | 0.696 | 0.961 | 0.955 | 0.688 | 0.819 | 0.884 | 0.565 | 0.882 | 0.977 | 0.776 | 0.878 |
| Image-level | MemAE | 0.454 | 0.946 | 0.611 | 0.63 | 0.967 | 0.954 | 0.694 | 0.831 | 0.891 | 0.537 | 0.883 | 0.992 | 0.972 | 0.793 | 0.871 |
| Image-level | AE+skip | 0.385 | 0.879 | 0.57 | 0.986 | 0.977 | 0.713 | 0.579 | 0.747 | 0.828 | 0.336 | 0.853 | 1 | 0.742 | 0.749 | 0.696 |
| Image-level | DAAD | 0.671 | 0.975 | 0.628 | 0.825 | 0.957 | 0.975 | 0.72 | 0.866 | 0.893 | 0.552 | 0.898 | 1 | 0.989 | 0.814 | 0.906 |
| Image-level | DAAD+ | 0.866 | 0.957 | 0.862 | 0.882 | 0.982 | 0.976 | 0.844 | 0.767 | 0.921 | 0.758 | 0.9 | 0.987 | 0.992 | 0.876 | 0.859 |
| Image-level | MMRR w/o ref. | 0.4166 | 0.981 | 0.8005 | 0.9015 | 0.9842 | 0.9458 | 0.8277 | 0.738 | 0.9157 | 0.7085 | 0.8862 | 0.5288 | 0.9816 | 0.8897 | 0.8629 |
| Image-level | MMRR | 0.496 | 0.9908 | 0.7993 | 0.7652 | 0.9316 | 0.9595 | 0.8639 | 0.7535 | 0.9107 | 0.8162 | 0.8775 | 0.66 | 0.9798 | 0.9162 | 0.8703 |

Table 7: Pixel-level and Image-level anomaly detection on MVTecAD dataset.

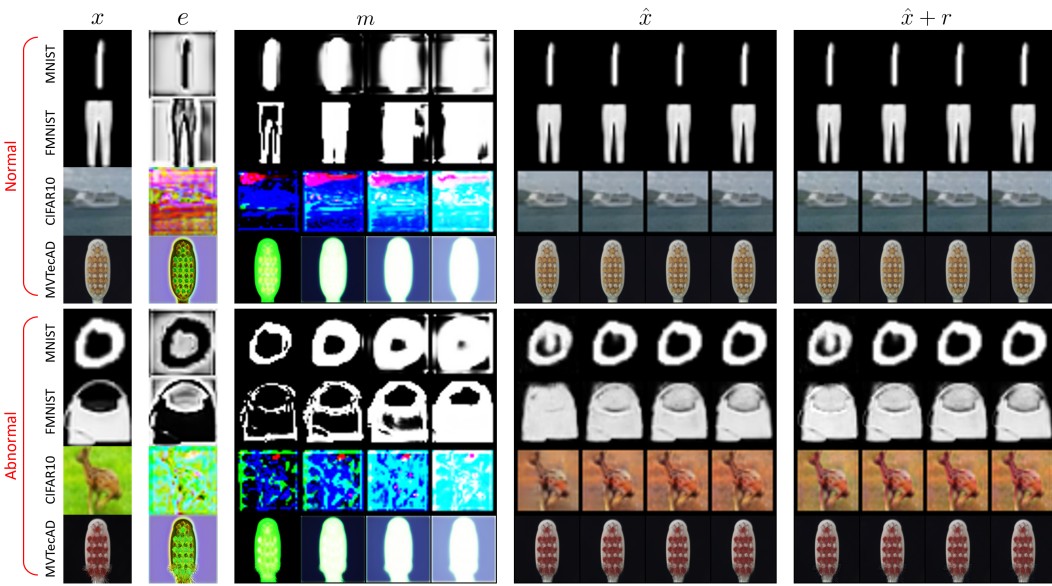

Figure 3: Qualitative results for normal and abnormal samples.

## 5 Conclusion

We proposed Multi-Level Masking and Restoration with Refinement (MMRR), which started from the motivation to perform anomaly detection through a series of processes of information limitation and restoration. The most noteworthy point of this study is that it presented the *hyperparameter sensitivity* problem for the first time, a problem that had been overlooked in existing anomaly detection studies. MMRR solved the *hyperparameter sensitivity* problem through ensemble at multiple masking levels with novel mask generation method. To empirically demonstrate the robustness to hyperparameter and prior knowledge-free properties of MMRR, we compared the performance as varying the number of masking level and augmentations. Additionally, we solved the problem of not considering the intrinsic complexity of data owing to the novel mask generation method through the refinement module, and achieved comparable performance on MNIST, FMNIST, CIFAR10, and MVTecAD datasets. However, since we have to forward several times for ensemble in multi-level masking, it has the disadvantage of being computationally expensive. We will go further here and try to find a lightweight anomaly detection method without suffering from *hyperparameter sensitivity* problems.

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
