| 0.539 | 0.96 | 0.751 | 0.476 | 0.63 | 0.909 | 0.732 | 0.786 | 0.976 | 0.88 | 0.885 | **0.979** | 0.971 | 0.906 | 0.68 |
| | VAE | 0.58 | 0.888 | 0.834 | 0.465 | 0.695 | 0.902 | 0.828 | 0.862 | 0.977 | 0.881 | 0.888 | 0.958 | 0.971 | 0.894 | 0.814 |
| | $\gamma - \mathrm{VAE}_g$ | **0.727** | **0.979** | **0.897** | 0.581 | **0.809** | **0.931** | 0.88 | 0.917 | **0.988** | **0.914** | **0.935** | 0.972 | **0.983** | **0.931** | 0.871 |
| | MMRR w/o ref. | 0.6733 | 0.8529 | 0.8599 | 0.7851 | 0.7911 | 0.8878 | **0.9117** | 0.898 | 0.8555 | 0.8648 | 0.9335 | 0.9074 | 0.9506 | 0.8865 | 0.8526 |
| | MMRR | 0.6561 | 0.8477 | 0.8405 | **0.7916** | 0.7858 | 0.889 | 0.8841 | **0.9179** | 0.9414 | 0.8197 | 0.9209 | 0.8924 | 0.9486 | 0.9038 | **0.8779** |
| Image-level | Ganomaly | 0.699 | 0.708 | 0.842 | 0.794 | 0.834 | 0.892 | 0.757 | 0.732 | 0.785 | 0.7 | 0.743 | 0.746 | 0.653 | 0.792 | 0.745 |
| | AE | 0.411 | 0.841 | 0.615 | 0.696 | 0.961 | 0.955 | 0.688 | 0.819 | 0.884 | 0.565 | 0.882 | 0.976 | 0.977 | 0.776 | 0.878 |
| | MemAE | 0.454 | 0.946 | 0.611 | 0.63 | 0.967 | 0.954 | 0.694 | 0.831 | 0.891 | 0.537 | 0.883 | 0.992 | 0.972 | 0.793 | 0.871 |
| | AE+skip | 0.385 | 0.879 | 0.57 | **0.986** | 0.977 | 0.713 | 0.579 | 0.747 | 0.828 | 0.336 | 0.853 | **1** | 0.742 | 0.749 | 0.696 |
| | DAAD | 0.671 | 0.975 | 0.628 | 0.825 | 0.957 | 0.975 | 0.72 | **0.866** | 0.893 | 0.552 | 0.898 | **1** | 0.989 | 0.814 | **0.906** |
| | DAAD+ | **0.866** | 0.957 | **0.862** | 0.882 | 0.982 | **0.976** | 0.844 | 0.767 | **0.921** | 0.758 | **0.9** | 0.987 | **0.992** | 0.876 | 0.859 |
| | MMRR w/o ref. | 0.4166 | 0.981 | 0.8005 | 0.9015 | **0.9842** | 0.9458 | 0.8277 | 0.738 | 0.9157 | 0.7085 | 0.8862 | 0.5288 | 0.9816 | 0.8897 | 0.8629 |
| | MMRR | 0.496 | **0.9908** | 0.7993 | 0.7652 | 0.9316 | 0.9595 | **0.8639** | 0.7535 | 0.9107 | **0.8162** | 0.8775 | 0.66 | 0.9798 | **0.9162** | 0.8703 |

Table 7: Pixel-level and Image-level anomaly detection on MVTecAD dataset.

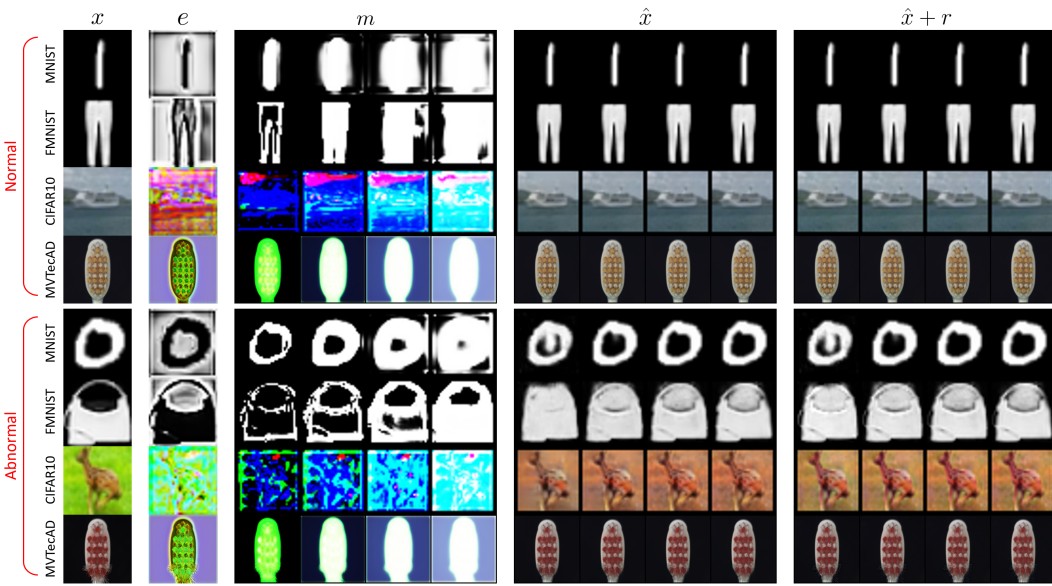

Figure 3: Qualitative results for normal and abnormal samples.

# 5 Conclusion

We proposed Multi-Level Masking and Restoration with Refinement (MMRR), which started from the motivation to perform anomaly detection through a series of processes of information limitation and restoration. The most noteworthy point of this study is that it presented the *hyperparameter sensitivity* problem for the first time, a problem that had been overlooked in existing anomaly detection studies. MMRR solved the *hyperparameter sensitivity* problem through ensemble at multiple masking levels with novel mask generation method. To empirically demonstrate the robustness to hyperparameter and prior knowledge-free properties of MMRR, we compared the performance as varying the number of masking level and augmentations. Additionally, we solved the problem of not considering the intrinsic complexity of data owing to the novel mask generation method through the refinement module, and achieved comparable performance on MNIST, FMNIST, CIFAR10, and MVTecAD datasets. However, since we have to forward several times for ensemble in multi-level masking, it has the disadvantage of being computationally expensive. We will go further here and try to find a lightweight anomaly detection method without suffering from *hyperparameter sensitivity* problems.

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

# A    Detailed Algorithm of Mask Generation Method

---
**Algorithm 1** Mask Generation Method (bisection method)

---
**Input:** Masking network output $f_M(x)$, masking level $\mu_m$
**Initialize:** $a = -1, c = 1$
**while** $sgn(\frac{1}{d}\sum_{i=1}^{d}\sigma(f_M(x)_i + a) - \mu_m) \neq sgn(\frac{1}{d}\sum_{i=1}^{d}\sigma(f_M(x)_i + c) - \mu_m)$ **do**     ▷ $sgn$ is sign function
$\quad a \leftarrow 2 \times a$
$\quad c \leftarrow 2 \times c$
**end while**
**for** $i = 1$ **to** $NMAX$ **do**                                    ▷ maximum iteration NMAX
$\quad b \leftarrow (a + c)/2$
$\quad$**if** $|\frac{1}{d}\sum_{i=1}^{d}\sigma(f_M(x)_i + b) - \mu_m| < TOL$ **then**                ▷ Tolerance value TOL
$\quad\quad m \leftarrow \sigma(f_M(x) + b)$
$\quad\quad$End
$\quad$**else if** $sgn(\frac{1}{d}\sum_{i=1}^{d}\sigma(f_M(x)_i + b) - \mu_m) = sgn(\frac{1}{d}\sum_{i=1}^{d}\sigma(f_M(x)_i + a) - \mu_m)$ **then**
$\quad\quad a \leftarrow b$
$\quad$**else if** $sgn(\frac{1}{d}\sum_{i=1}^{d}\sigma(f_M(x)_i + c) - \mu_m) = sgn(\frac{1}{d}\sum_{i=1}^{d}\sigma(f_M(x)_i + b) - \mu_m)$ **then**
$\quad\quad c \leftarrow b$
$\quad$**end if**
**end for**
**Output:** Mask $m$

---

# B    Derivative of Mask Generation Method

We assume that the bias $b$ that makes the average value of mask $m$ to the masking level $\mu_m$ is found

well through the bisection method($\mu_m = \frac{1}{d}\sum_{i=1}^{d}\sigma(f_M(x)_i + b)$, where $f_M(x)$ is masking network

output and $d$ is number of data dimensions.) as can be seen Alg. 1. We obtained the gradient $\frac{\partial\mathcal{L}}{\partial f_M(x)_i}$

under this assumption.

$$\frac{\partial\mathcal{L}}{\partial f_M(x)_i} = \sum_{\forall j}\frac{\partial\mathcal{L}}{\partial m_j}\frac{\partial m_j}{\partial f_M(x)_i}$$

Since mask $m_j = \sigma(f_M(x)_j + b)$,

$$\frac{\partial m_j}{\partial f_M(x)_i} = m_j(1 - m_j)(\frac{\partial f_M(x)_j}{\partial f_M(x)_i} + \frac{\partial b}{\partial f_M(x)_i})$$

$$= m_j(1 - m_j)(\delta(i,j) + \frac{\partial b}{\partial f_M(x)_i}), \qquad\qquad \delta(i,j) = \begin{cases} 1, & \text{if } i = j \\ 0, & \text{otherwise} \end{cases}.$$

Since $\mu_m = \frac{1}{d}\sum_{i=1}^{d}\sigma(f_M(x)_k + b)$ and $\frac{\partial\mu_m}{\partial f_M(x)_i} = 0$,

$$\frac{\partial\mu_m}{\partial f_M(x)_i} = \frac{1}{d}\sum_{\forall k}m_k(1 - m_k)(\frac{\partial f_M(x)_k}{\partial f_M(x)_i} + \frac{\partial b}{\partial f_M(x)_i})$$

$$0 = \sum_{\forall k}m_k(1 - m_k)(\delta(i,k) + \frac{\partial b}{\partial f_M(x)_i})$$

$$\frac{\partial b}{\partial f_M(x)_i} = \frac{-m_i(1 - m_i)}{\sum_{\forall k}m_k(1 - m_k)}$$

Finally,

$$\frac{\partial m_j}{\partial f_M(x)_i} = m_j(1 - m_j)(\delta(i,j) - \frac{m_i(1 - m_i)}{\sum_{\forall k}m_k(1 - m_k)})$$

$$\frac{\partial\mathcal{L}}{\partial f_M(x)_i} = \sum_{\forall j}\frac{\partial\mathcal{L}}{\partial m_j}m_j(1 - m_j)(\delta(i,j) - \frac{m_i(1 - m_i)}{\sum_{\forall k}m_k(1 - m_k)})$$

# C Detailed Experimental Results

Detailed settings of experiments were described in Section 4. All experiements were conducted with 2080ti and TITAN Xp GPUs.

| MNIST | OC-SVM | KDE | AnoGAN | DSVDD | OC-GAN | CAVGA | MMRR w/o ref. | MMRR |
|---|---|---|---|---|---|---|---|---|
| 0 | 0.988 | 0.885 | 0.966 | 0.98 | **0.998** | 0.994 | 0.9857 | $0.9941 \pm 0.0007$ |
| 1 | **0.999** | 0.996 | 0.992 | 0.997 | **0.999** | 0.997 | **0.999** | $0.9982 \pm 0.0005$ |
| 2 | 0.902 | 0.71 | 0.85 | 0.917 | 0.942 | **0.989** | 0.8981 | $0.94 \pm 0.0053$ |
| 3 | 0.95 | 0.693 | 0.887 | 0.919 | 0.963 | **0.983** | 0.9246 | $0.955 \pm 0.0086$ |
| 4 | 0.955 | 0.844 | 0.894 | 0.949 | 0.975 | **0.977** | 0.9309 | $0.9352 \pm 0.0051$ |
| 5 | 0.968 | 0.776 | 0.883 | 0.885 | **0.98** | 0.968 | 0.9173 | $0.971 \pm 0.006$ |
| 6 | 0.978 | 0.861 | 0.947 | 0.983 | **0.991** | 0.988 | 0.9765 | $0.989 \pm 0.0017$ |
| 7 | 0.965 | 0.884 | 0.935 | 0.946 | 0.981 | **0.986** | 0.9539 | $0.966 \pm 0.0012$ |
| 8 | 0.853 | 0.669 | 0.849 | 0.939 | 0.939 | **0.988** | 0.906 | $0.945 \pm 0.0107$ |
| 9 | 0.955 | 0.825 | 0.924 | 0.965 | 0.981 | **0.991** | 0.9511 | $0.98 \pm 0.0041$ |

Table 8: Image-level AUROC for one-vs-all anomaly detection on MNIST with error bar.

| CIFAR10 | OC-SVM | KDE | AnoGAN | DSVDD | OC-GAN | $\gamma$-VAE | CAVGA | DROCC | MMRR w/o ref | MMRR |
|---|---|---|---|---|---|---|---|---|---|---|
| Airplane | 0.63 | 0.658 | 0.671 | 0.617 | 0.757 | 0.702 | 0.653 | **0.817** | 0.7778 | $0.7965 \pm 0.0095$ |
| Automobile | 0.44 | 0.52 | 0.547 | 0.659 | 0.531 | 0.663 | **0.784** | 0.767 | 0.6065 | $0.7377 \pm 0.0064$ |
| Bird | 0.649 | 0.657 | 0.529 | 0.508 | 0.64 | 0.68 | **0.761** | 0.667 | 0.6926 | $0.7024 \pm 0.0099$ |
| Cat | 0.487 | 0.497 | 0.545 | 0.591 | 0.62 | 0.713 | **0.747** | 0.671 | 0.6076 | $0.6595 \pm 0.0074$ |
| Deer | 0.735 | 0.727 | 0.651 | 0.609 | 0.723 | 0.77 | 0.775 | 0.736 | 0.7638 | $\mathbf{0.7817 \pm 0.0109}$ |
| Dog | 0.5 | 0.496 | 0.603 | 0.657 | 0.62 | 0.689 | 0.552 | **0.744** | 0.6143 | $0.6739 \pm 0.0173$ |
| Frog | 0.725 | 0.758 | 0.585 | 0.677 | 0.723 | 0.805 | **0.813** | 0.744 | 0.6966 | $0.7641 \pm 0.0137$ |
| Horse | 0.533 | 0.564 | 0.625 | 0.673 | 0.575 | 0.588 | **0.745** | 0.714 | 0.626 | $0.7037 \pm 0.009$ |
| Ship | 0.649 | 0.68 | 0.758 | 0.759 | **0.82** | 0.813 | 0.801 | 0.800 | 0.7878 | $0.8181 \pm 0.0134$ |
| Truck | 0.508 | 0.54 | 0.665 | 0.731 | 0.554 | 0.744 | 0.741 | **0.762** | 0.6229 | $0.7325 \pm 0.0149$ |

Table 9: Image-level AUROC for one-vs-all anomaly detection on CIFAR10 with error bar.

| | Method | carpet | grid | leather | tile | wood | bottle | cable | capsule | hazelnut | metalnut | pill | screw | toothbrush | transistor | zipper |
|---|---|---|---|---|---|---|---|---|---|---|---|---|---|---|---|---|
| Pixel-level | AE | 0.539 | 0.96 | 0.751 | 0.476 | 0.63 | 0.909 | 0.732 | 0.786 | 0.976 | 0.88 | 0.885 | **0.979** | 0.971 | 0.906 | 0.68 |
| | VAE | 0.58 | 0.888 | 0.834 | 0.465 | 0.695 | 0.902 | 0.828 | 0.862 | 0.977 | 0.881 | 0.888 | 0.958 | 0.971 | 0.894 | 0.814 |
| | $\gamma - VAE_g$ | **0.727** | **0.979** | **0.897** | 0.581 | **0.809** | **0.931** | 0.88 | 0.917 | **0.988** | **0.914** | **0.935** | 0.972 | **0.983** | **0.931** | 0.871 |
| | MMRR w/o ref. | 0.6733 | 0.8529 | 0.8599 | 0.7851 | 0.7911 | 0.8878 | **0.9117** | 0.898 | 0.8555 | 0.8648 | 0.9335 | 0.9074 | 0.9506 | 0.8865 | 0.8526 |
| | MMRR | $0.6561 \pm 0.0416$ | $0.8477 \pm 0.012$ | $0.8405 \pm 0.093$ | $\mathbf{0.7916 \pm 0.044}$ | $0.7858 \pm 0.0284$ | $0.889 \pm 0.0055$ | $0.8841 \pm 0.007$ | $\mathbf{0.9179 \pm 0.022}$ | $0.9414 \pm 0.0107$ | $0.8197 \pm 0.0133$ | $0.9209 \pm 0.0285$ | $0.8924 \pm 0.0157$ | $0.9486 \pm 0.003$ | $0.9038 \pm 0.0114$ | $\mathbf{0.8779 \pm 0.0054}$ |
| Image-level | Ganomaly | 0.699 | 0.708 | 0.842 | 0.794 | 0.834 | 0.892 | 0.757 | 0.732 | 0.785 | 0.7 | 0.743 | 0.746 | 0.653 | 0.792 | 0.745 |
| | AE | 0.411 | 0.841 | 0.615 | 0.696 | 0.961 | 0.955 | 0.688 | 0.819 | 0.884 | 0.565 | 0.882 | 0.956 | 0.977 | 0.776 | 0.878 |
| | MemAE | 0.454 | 0.946 | 0.611 | 0.63 | 0.967 | 0.954 | 0.694 | 0.831 | 0.891 | 0.537 | 0.883 | 0.992 | 0.972 | 0.793 | 0.871 |
| | AE+skip | 0.385 | 0.879 | 0.57 | **0.986** | 0.977 | 0.975 | 0.713 | 0.747 | 0.828 | 0.336 | 0.853 | **1** | 0.742 | 0.749 | 0.696 |
| | DAAD | 0.671 | 0.975 | 0.628 | 0.825 | 0.957 | 0.975 | 0.72 | **0.866** | 0.893 | 0.552 | 0.898 | **1** | 0.989 | 0.814 | **0.906** |
| | DAAD+ | **0.866** | 0.957 | **0.862** | 0.882 | 0.982 | **0.976** | 0.844 | 0.767 | **0.921** | 0.758 | 0.9 | 0.987 | **0.992** | 0.876 | 0.859 |
| | MMRR w/o ref. | 0.4166 | 0.981 | 0.8005 | 0.9015 | **0.9842** | 0.9458 | 0.8277 | 0.738 | 0.9157 | 0.7085 | 0.8862 | 0.5288 | 0.9816 | 0.8897 | 0.8629 |
| | MMRR | $0.496 \pm 0.029$ | $\mathbf{0.9908 \pm 0.0155}$ | $0.7993 \pm 0.0736$ | $0.7652 \pm 0.0628$ | $0.9316 \pm 0.0583$ | $0.9595 \pm 0.0058$ | $\mathbf{0.8639 \pm 0.0312}$ | $0.7535 \pm 0.0289$ | $0.9107 \pm 0.0351$ | $\mathbf{0.8162 \pm 0.0488}$ | $0.8775 \pm 0.0229$ | $0.66 \pm 0.0816$ | $0.9798 \pm 0.0249$ | $\mathbf{0.9162 \pm 0.0269}$ | $0.8703 \pm 0.0708$ |

Table 10: Pixel-level and Image-level anomaly detection on MVTecAD dataset with mvtec with error bar.

# D Qualitative Results

In this section, we will visualize data $x$, embedding $e$, mask $m$ for $L$ masking levels, reconstructed output $\hat{x}$, and refined output $\hat{x} + r$ for all datasets(MNIST, FMNIST, CIFAR10, MVTecAD) we used. We will show $L = 15$ masking levels for MVTecAD dataset and $L = 60$ masking levels for other datasets.

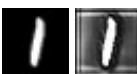

Figure 4: Normal sample $x$ and corresponding $e$ on MNIST dataset

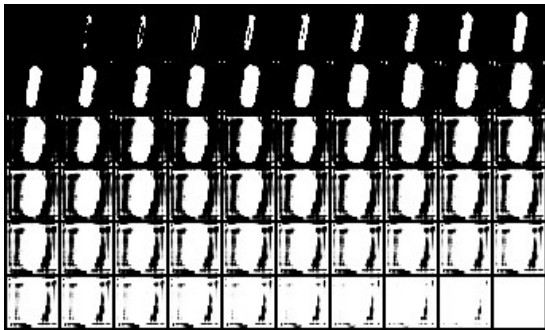

Figure 5: Mask $m$ of normal sample $x$ with $L = 60$ different masking levels $\mu_m$ on MNIST dataset

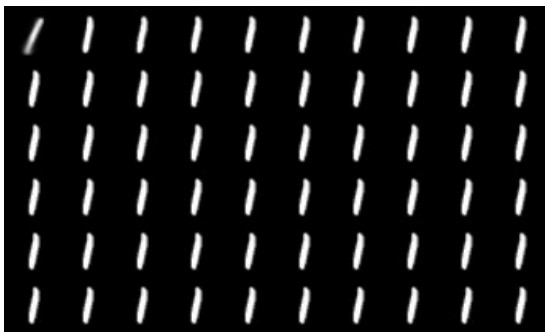

Figure 6: Reconstructed output $\hat{x}$ of normal sample $x$ on MNIST dataset

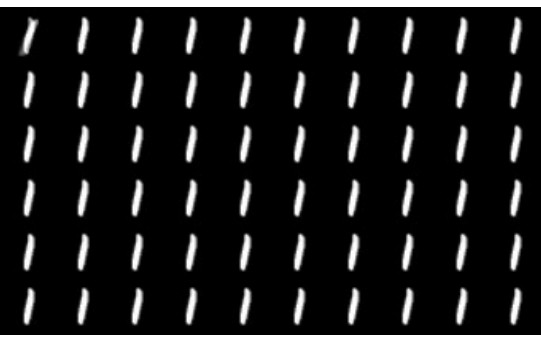

Figure 7: Refined output $\hat{x} + r$ of normal sample $x$ on MNIST dataset

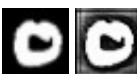

Figure 8: Abnormal sample $x$ and corresponding $e$ on MNIST dataset

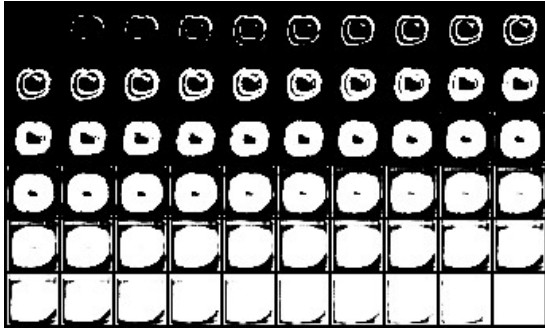

Figure 9: Mask $m$ of abnormal sample $x$ with $L = 60$ different masking levels $\mu_m$ on MNIST dataset

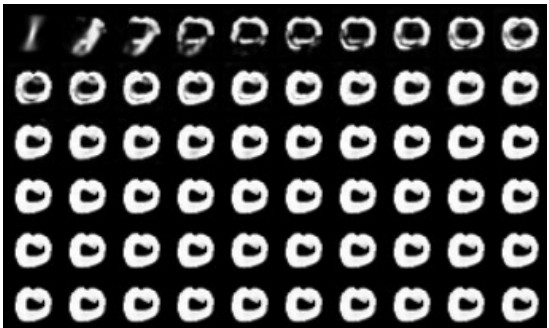

Figure 10: Reconstructed output $\hat{x}$ of abnormal sample $x$ on MNIST dataset

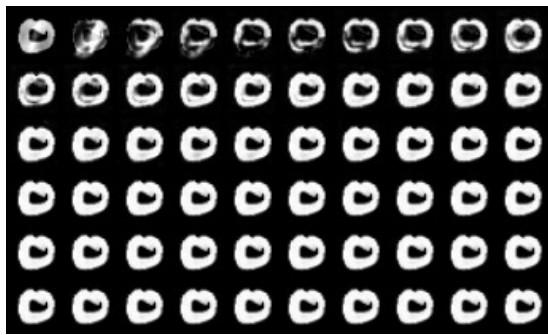

Figure 11: Refined output $\hat{x} + r$ of abnormal sample $x$ on MNIST dataset

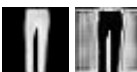

Figure 12: Normal sample $x$ and corresponding $e$ on FMNIST dataset

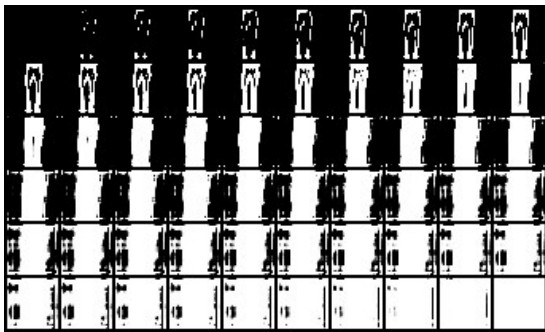

Figure 13: Mask $m$ of normal sample $x$ with $L = 60$ different masking levels $\mu_m$ on FMNIST dataset

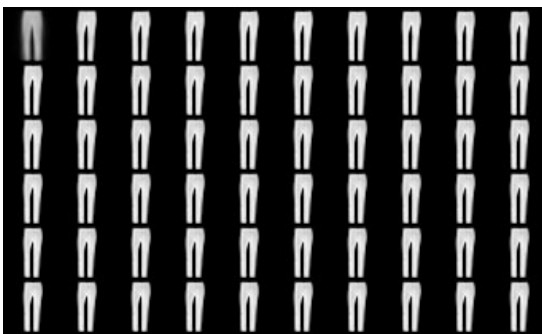

Figure 14: Reconstructed output $\hat{x}$ of normal sample $x$ on FMNIST dataset

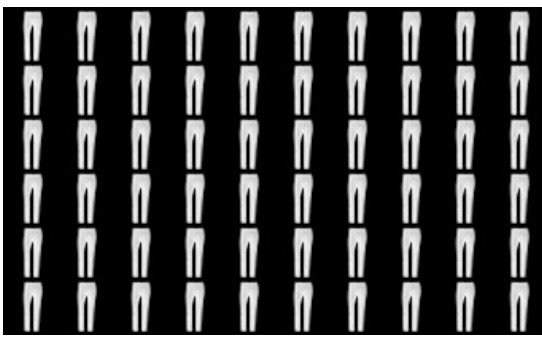

Figure 15: Refined output $\hat{x} + r$ of normal sample $x$ on FMNIST dataset

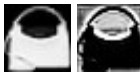

Figure 16: Abnormal sample $x$ and corresponding $e$ on FMNIST dataset

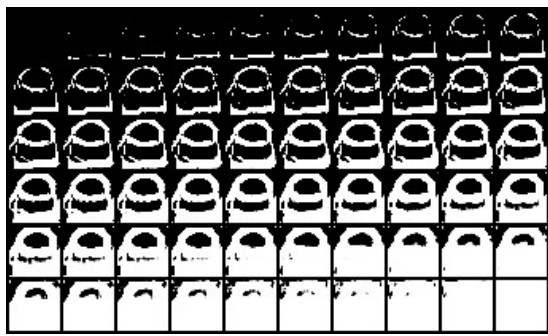

Figure 17: Mask $m$ of abnormal sample $x$ with $L = 60$ different masking levels $\mu_m$ on FMNIST dataset

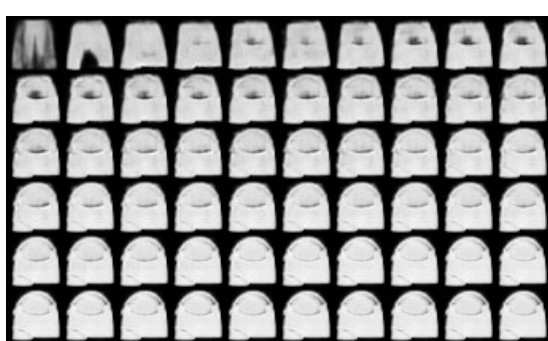

Figure 18: Reconstructed output $\hat{x}$ of abnormal sample $x$ on FMNIST dataset

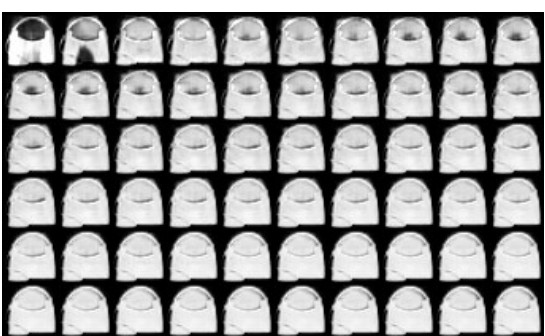

Figure 19: Refined output $\hat{x} + r$ of abnormal sample $x$ on FMNIST dataset

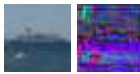

Figure 20: Normal sample $x$ and corresponding $e$ on CIFAR10 dataset

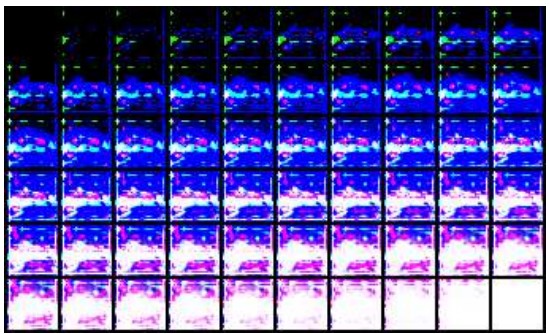

Figure 21: Mask $m$ of normal sample $x$ with $L = 60$ different masking levels $\mu_m$ on CIFAR10 dataset

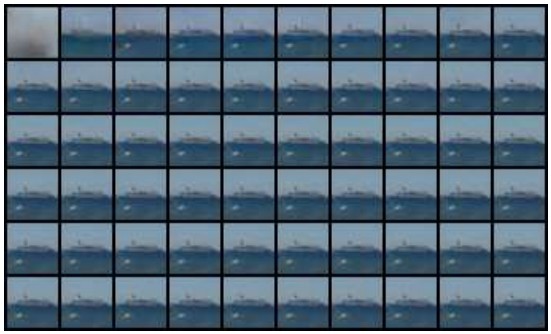

Figure 22: Reconstructed output $\hat{x}$ of normal sample $x$ on CIFAR10 dataset

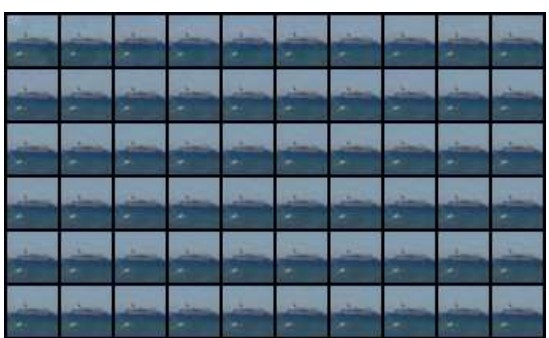

Figure 23: Refined output $\hat{x} + r$ of normal sample $x$ on CIFAR10 dataset

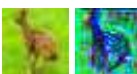

Figure 24: Abnormal sample $x$ and corresponding $e$ on CIFAR10 dataset

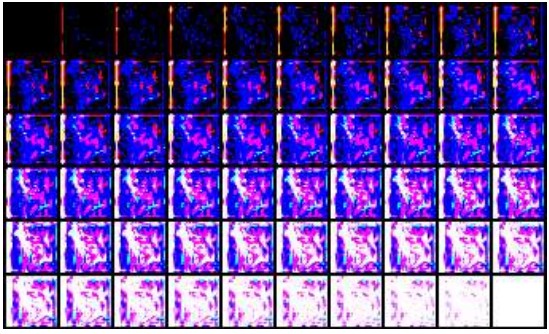

Figure 25: Mask $m$ of abnormal sample $x$ with $L = 60$ different masking levels $\mu_m$ on CIFAR10 dataset

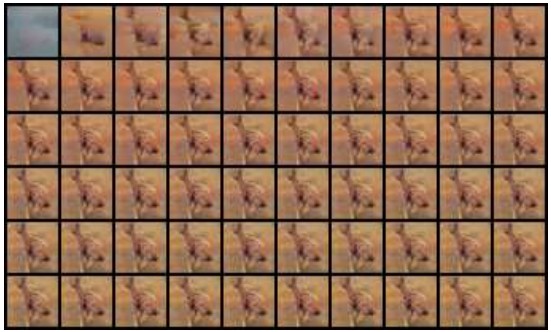

Figure 26: Reconstructed output $\hat{x}$ of abnormal sample $x$ on CIFAR10 dataset

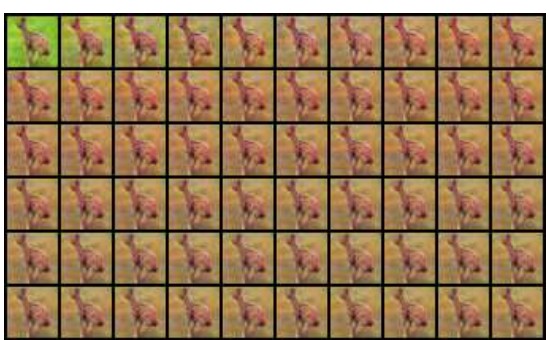

Figure 27: Refined output $\hat{x} + r$ of abnormal sample $x$ on CIFAR10 dataset

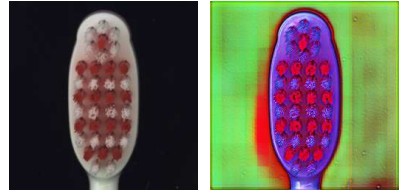

Figure 28: Normal sample $x$ and corresponing $e$ on MVTecAD dataset

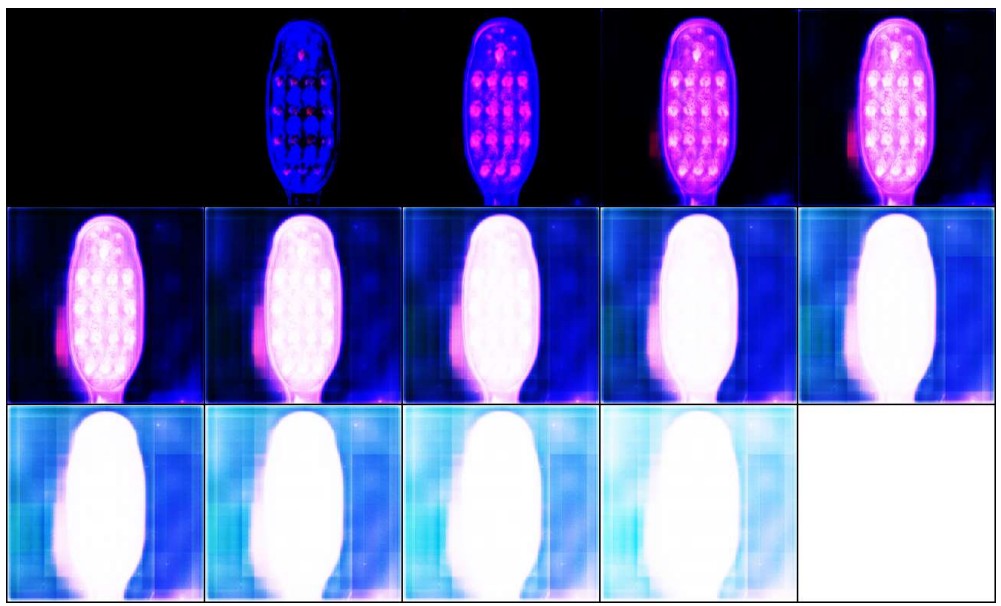

Figure 29: Mask $m$ of normal sample $x$ with $L = 15$ different masking levels $\mu_m$ on MVTecAD dataset

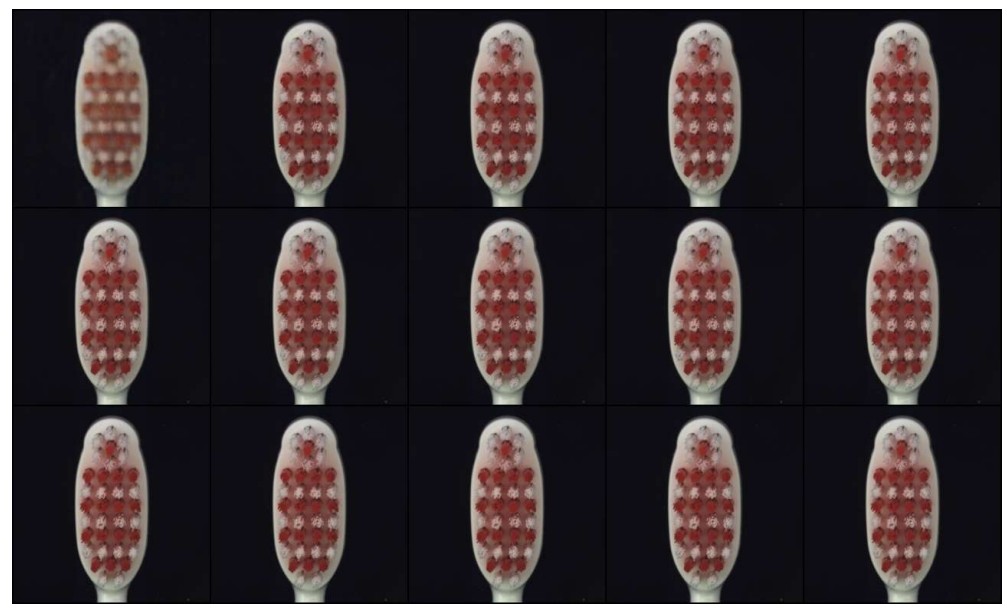

Figure 30: Reconstructed output $\hat{x}$ of normal sample $x$ on MVTecAD dataset

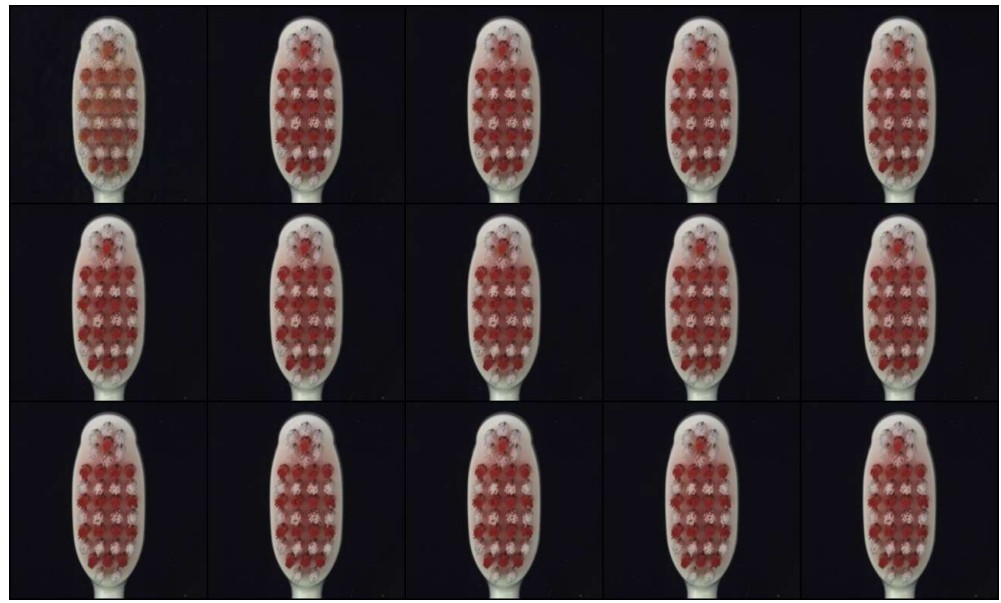

Figure 31: Refined output $\hat{x} + r$ of normal sample $x$ on MVTecAD dataset

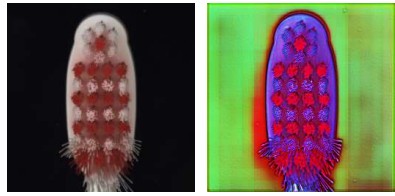

Figure 32: Abnormal sample $x$ and corresponding $e$ on MVTecAD dataset

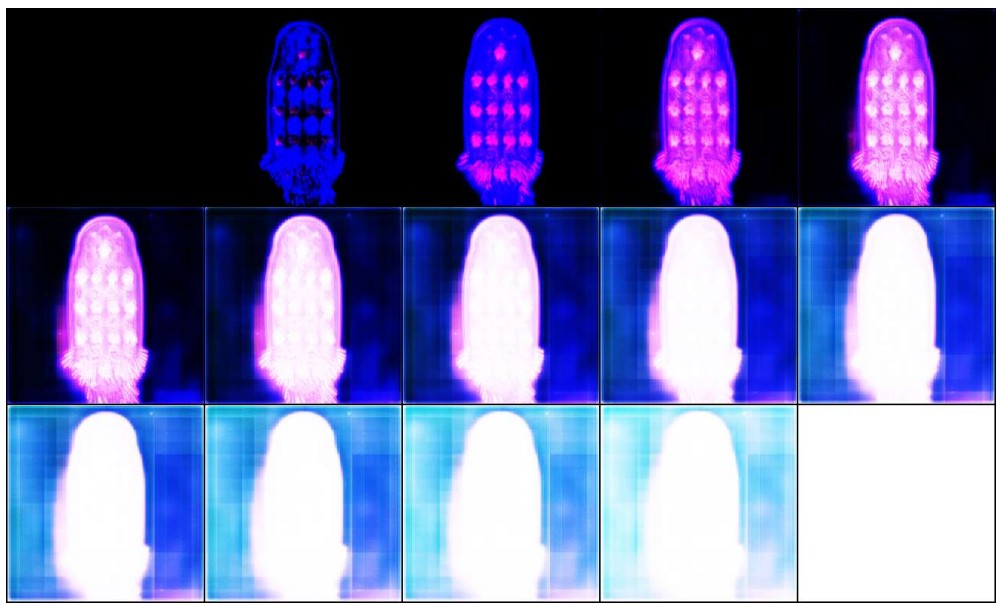

Figure 33: Mask $m$ of abnormal sample $x$ with $L = 15$ different masking levels $\mu_m$ on MVTecAD dataset

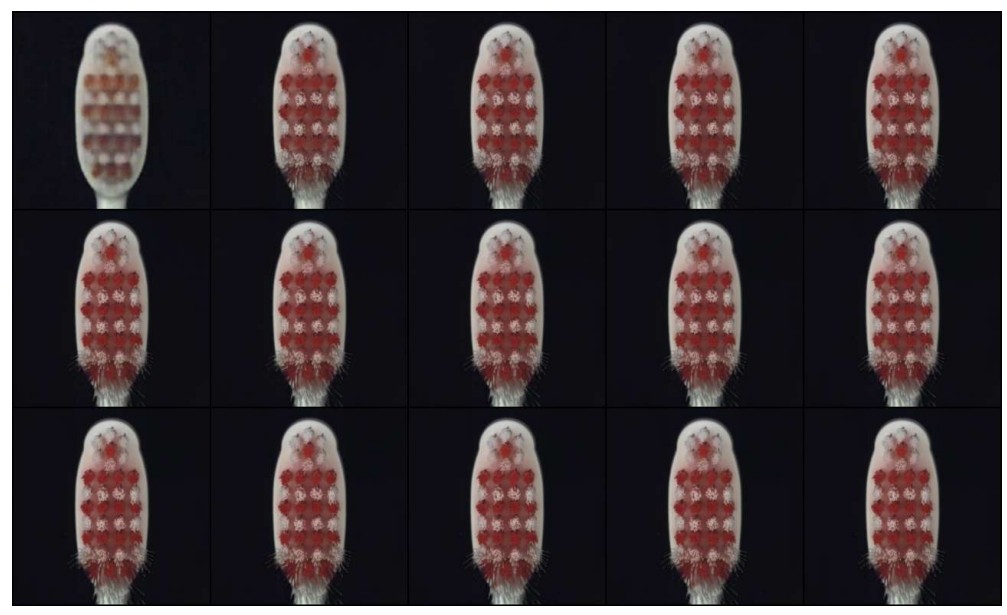

Figure 34: Reconstructed output $\hat{x}$ of abnormal sample $x$ on MVTecAD dataset

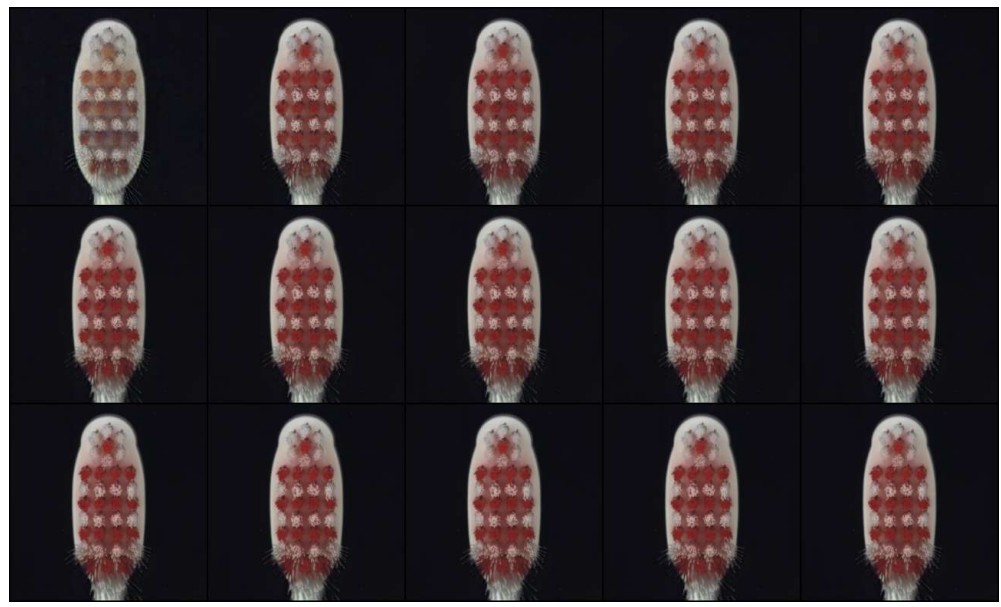

Figure 35: Refined output $\hat{x} + r$ of abnormal sample $x$ on MVTecAD dataset