# OpenReview forum: "MMRR: Unsupervised Anomaly Detection through Multi-Level Masking and Restoration with Refinement"
_NeurIPS.cc/2022/Conference — NeurIPS 2022 Submitted_

### Official Review · Reviewer_7z6Y · 2022-06-30

**Rating:** 5
**Confidence:** 4
**Soundness:** 2 fair
**Presentation:** 2 fair
**Contribution:** 2 fair

**Summary:**

This paper introduces a reconstruction-based anomaly detection algorithm. The algorithm is based on autoencoders, and noise is added in the form of masking of one of the embedding layers, which then has to be reconstructed to match the original input; input-specific masks are computed using a masking network, with a given average masking level. A further refinement network is implemented to correct the residual error from the reconstruction. At test time, predictions from various masking levels are ensembled and the reconstruction error post refinement is used as the anomaly score. Experiments performed on benchmark image anomaly datasets show competitive results with recent methods.


**Questions:**

- Does the root-finding algorithm find a solution to the equation in practice? Is the assumption used to derive the gradient in Eq (2) satisfied?

- How does the method perform on tabular or other non-image datasets (e.g. ODDS datasets)?

- How does the method perform relative to other masking-based autoencoders?

- What is the performance with a uniform mask (see above)?


**Limitations:**

While I agree that the proposed method does eliminate the choice of the masking level hyperparameter, the autoencoder architecture to use and training hyperparameters still have to be tuned for a specific dataset, so I think it is a bit of an overreach to claim that the proposed method solves the tuning issue. To be fair this is an issue with many other methods as well.


**Strengths And Weaknesses:**

Overall, the paper presents a promising approach to anomaly detection but some aspects of the algorithm are not well motivated or validated, and the evaluation could be improved. Detailed comments follow.

**Originality:**

This work extends a long line of reconstruction-based anomaly detection methods using autoencoders, with recent ones including noise and even masking but at the input layers (see references below). The key difference here is in implementing the masking at the inner embedding layers, and the use of a refinement network to further improve the reconstructions. From this perspective, the proposed method seems incremental.

- One-Class Learned Encoder-Decoder Network with Adversarial Context Masking for Novelty Detection. WACV 2022.
- Self-Supervised Masking for Unsupervised Anomaly Detection and Localization. IEEE Transactions on Multimedia 2022.


**Quality:**

Overall the approach and evaluation are reasonable, though there are several aspects that need to be addressed:

- The paper does not motivate well the need to mask embeddings versus input layers (I did not understand the justification provided on L142), nor are comparisons provided to input masking autoencoder methods provided which could help to motivate the method.

- The paper differentiates through the mask generation step by assuming that the average masking level condition is satisfied (Line 169). However, it is not shown whether this condition is indeed satisfied in the paper - this should be verified to ensure that the method is technically sound.

- Experiments in the paper are limited to images, though the framework appears to be generic. It would be informative to include tabular datasets (e.g. from ODDS - http://odds.cs.stonybrook.edu/) so the generality of the masking strategy can be assessed. Moreover, "side-information" is not usually available for such data and would be better able to demonstrate the practical utility of the proposed method.

- Given that the mask is in [0,1], the ablation study (L326) should also include masks with entries uniformly generated over [0,1].


**Clarity:**
- The paper is largely clear but there are several typos scattered across the paper (e.g. L23 real-word, L154 successfullt, L158 owingto, L173 non-differential).

- It would be helpful to provide the percentage of anomalies in each dataset.

- Just a suggestion: it may be clearer to revise the definition of masking to $\tilde{e} = e \odot (1-m) + \epsilon \odot m$, such that a higher m represents a higher masking level in that more of the input is removed.


**Significance:**

Overall, while the method is interesting and has some novel elements, the significance of the results is not clear:

- In terms of the method, the refinement module is independent of the masking approach, and could also be applied to other autoencoder methods; the notion of masking is somewhat interesting and broadly applicable, but the lack of good performance makes its significance unclear.

- In terms of performance, given that other better performing methods exist on the image datasets evaluated on, it is unclear if the method will be practically useful as other training/model hyperparameters still need to be tuned in practice. As mentioned above, it would be useful to demonstrate good performance on other datasets for which side-information is not available.

- The claim that this is the first paper presenting the "hyperparameter sensitity" is not quite true, as this has been reported in the following workshop paper.

The Effect of Hyperparameter Tuning on the Comparative Evaluation of Unsupervised Anomaly Detection Methods. 6th Outlier Detection and Description Workshop @ KDD 2021.


**Other comments:**

- Given that some results are pretty close, it would be informative to provide results from multiple runs along with standard deviations.

- In addition, providing results with the more appropriate AUPRC metric for anomaly detection due to the class imbalance would be informative.

** Post rebuttal update: **
As described in the comment below, the response has addressed some of my key concerns regarding comparison to other masked-based anomaly detection methods and I am raising my score to a 5.

---

> ### Author Response · Authors · 2022-08-01
> **Rebuttal for [R3]**
>
> # 1. Comparison with other mask-based restoration methods
> Common rebuttal for [R1, R3]: Comparison with other mask-based restoration methods
> # 2. Performance in terms of AUPRC
> Common rebuttal for [R2, R3]: Performance in terms of AUPRC
> # 3. why do we mask embeddings
> There are two reasons.(ablation study can be shown in Table 2.)
> First, as the distribution of noise and data used for masking is similar, it will be difficult to distinguish between noise and data. And this will lead to difficulties in restoration. Therefore, we tried to ensure that all data can be equally distinguished from noise through the embedding process.
> Second, like the assumptions of other anomaly detection studies using autoencoder, the embedding generated through the embedding process was expected to be helpful in anomaly detection by learning the manifold for normal data.
> # 4. Does the root-finding algorithm find a solution to the equation in practice?
> Sigmoid function is a monotonically increasing function with respect to the bias value, so we can always find the bias that satisfies the equation. If you look at the Algorithm 1 in supplementary material, you can find out the detailed mask generation method.
> # 5. Non image dataset experiments
> Experiments were conducted with the thyroid and arrhythmia datasets in the same settings as [1].
> Experimental results show that MMRR (without refinement / with refinement) performs much better than SOTA method DROCC[2], on  non-image tabular dataset.
> In the case of the network structure, an autoencoder consisting of only fully connected layers was used, and a skip connection was applied between features of the same size as in UNet.
> |       | Thyroid (F1) | Arrythmia (F1) |
> |-------|--------------|----------------|
> | DROCC | 0.78         | 0.69           |
> | MMRR  | 0.685/**0.8596** | 0.7244/**0.7427**  |
>
> # 6. Some hyperparameters still need to be adjusted.
> Common rebuttal for [R2, R3]: Is the hyperparameter sensitivity problem really important and is it resolved?
> # 7. Provide results from multiple runs along with standard deviations.
> Tables 8, 9, and 10 of the supplementary material describe the std for multiple runs.
> # 8. Performance with a uniform mask
> We proved the effectiveness of our masking method by comparing it with the bernoulli sampled mask and the constant mask. A mask sampled from uniform noise may also be one of the masks we can easily think of.
> However, the reason we did not use the uniform mask for the ablation study is that the mask mean cannot be adjusted to 0~1  unlike the mask of MMRR, bernoulli mask, and constant mask. Therefore, I thought that it would not be a fair comparison. However, an additional experiment was conducted on CIFAR10 and the following is the experimental result(AUROC) using a uniform mask.
> |     | w/o ref | w/ ref  |
> |------|---------|---------|
> | 0    | 0.7491  | 0.7461  |
> | 1    | 0.445   | 0.5747  |
> | 2    | 0.6799  | 0.6664  |
> | 3    | 0.6084  | 0.6462  |
> | 4    | 0.7107  | 0.7554  |
> | 5    | 0.6149  | 0.6283  |
> | 6    | 0.603   | 0.6237  |
> | 7    | 0.5531  | 0.6061  |
> | 8    | 0.7505  | 0.7599  |
> | 9    | 0.4411  | 0.5764  |
> | MEAN | 0.61557 | 0.65832 |
>
>
> ## Reference
> [1] Zong, Bo, et al. "Deep autoencoding gaussian mixture model for unsupervised anomaly detection." International conference on learning representations. 2018.\
> [2] Goyal, Sachin, et al. "DROCC: Deep robust one-class classification." International Conference on Machine Learning. PMLR, 2020.

---

> ### Author Response · Authors · 2022-08-09
> **The discussion period closes in about 12 hours**
>
> Today is the end of the discussion deadline. Could you please go over our rebuttal and check the responses? We believe that we have addressed all your concerns and that including these discussions will further strengthen our paper. We hope you reflect this in your final review and the score. We thank you again for your time and efforts in reviewing our paper.

---

> ### Comment · Reviewer_7z6Y · 2022-08-09
> **Response to rebuttal**
>
> Thanks for the responses.
>
> Regarding the request for AUPRC numbers - I would have liked to see a comparison of AUPRC numbers with the other methods, not just those for the proposed method.
>
> On the comparison to other masked-based methods - the rebuttal has addressed my concern.
>
> On the response to the hyper-parameter sensitivity problem, I do agree with R2 that in practice all methods are likely to be used by empirically setting a threshold with small numbers of anomalies, but I also appreciate the perspective that it could be interesting to study the strict setting where such labeled anomalies are not available. With respect to the response on model design, a greater variation in architectures would have to be explore to more definitively make the claim that architecture and training hyperparameters do not make a big difference; in particular, U-Nets would not have been my choice for such networks given that they were designed for segmentation and conventional ResNets or transformers would have come to mind first.
>
> On results on non-image data: the presented results on tabular datasets are promising and are a good start. I also agree with R2 underlying point that in real situations, one will pick the method based on prior knowledge of the proposed anomalies. Given that such knowledge is available in the datasets of choice, again I suggest finding other datasets where this is not available to more convincingly demonstrate the utility of the approach.
>
> Overall, the response has addressed some of my concerns and I will raise my score. That said, a convincing demonstration of the practical utility of the method on other datasets including comparisons with the other masked-based reconstruction methods will make for a more convincing paper.

---

> > ### Author Response · Authors · 2022-08-09
> > **Response for [R3] comment**
> >
> > Thank you for suggesting meaningful improvements such as more variation comparison in model architecture, comparison with other masking based methods, and experiments on datasets where methods using prior knowledge are not applied.
> > Additionally, comparison experiments with methods using prior knowledge on non-image datasets can be seen in response to [R2]. Of course, in order to apply the concept of geometric transformation to non-image datasets, a universally available affine transformation was used instead of the specialized transformations (rotation, movement, flip, etc.) used in image data.

---

### Official Review · Reviewer_rzLV · 2022-07-04

**Rating:** 3
**Confidence:** 4
**Soundness:** 2 fair
**Presentation:** 3 good
**Contribution:** 2 fair

**Summary:**

This work targeted the unsupervised anomaly detection problem and proposed a reconstruction-based method. Namely, by masking out features of the input data, and adding also noise to the features, the reconstruction and refinement network are only trained on the normal data to achieve small reconstruction errors. Therefore, any anomaly would generally suffer from high reconstruction errors. The evaluation is carried out in the one-vs-rest setup of the multi-class datasets MNIST, FMNIST, CIFAR10, and the defect detection benchmark MVTecAD. These benchmarks cover both semantic and non-semantic anomaly patterns, showing the covered use scenarios of the method.

The method's gain over SoTAs is quite diverse on the benchmarks. The authors argued an important advantage of the proposed method is its less sensitivity to hyperparameters, e.g., through an ensemble of different levels of masking, and the performance generally increases with more levels and is saturated at a certain level.

**Questions:**

1. AUPR evaluation: for an imbalanced evaluation set (the ratio of normal and abnormal samples is not balanced, AUPR is a more insightful metric to evaluate than AUROC.

2. Analysis of the learned masking patterns: how they behave differently on different datasets. For instance, in the visual example Fig. 3, while the model is trained on ship of CIFAR10, it can still reconstruct the deer class. This is a known observation of reconstruction-based methods. The proposed method does not seem to provide a different observation. The reconstruction difference is mainly on the local pixels. This leads to the question if the network really exploits the semantic structure to mask and reconstruct. This may explain why the performance on CIFAR10 is quite diverse across different classes.

3. On the MVTec benchmark, there are more competitive baselines, e.g., from https://paperswithcode.com/sota/anomaly-detection-on-mvtec-ad.

4. On the F/MNNIST and CIFAR10, the authors mainly evaluated on the one-vs-rest setting. how this approach can handle multi-class settings, namely, the inlier dataset is multi-class.

5. How well this approach can scale up to more diverse and higher-resolution datasets?

6. More discussion or empirical analysis of the effectiveness of feature masking. For instance, on which feature dimension, how does it change over different input resolutions or types of anomaly patterns, the importance of adding noise vs. just putting zeros, why the learned mask does not select a shortcut, e.g., masking out every n pixels (like a downsampling) to achieve good reconstruction.

**Limitations:**

The limitation discussion of this work is primarily focused on cases where the method is performing on a par or worse than baselines, and also on the ablation study part. However, as provided in the previous feedback sessions (weakness and questions), I think it is still unclear on which conditions/reasons the model is particularly good and in which scenarios it is expected to be underperforming. This is very important as so far the empirical results are quite diverse in terms of gains over SoTAs.

**Strengths And Weaknesses:**

Reconstruction is one type of method to address anomaly detection. It is an unsupervised approach, thus requiring little knowledge of anomalies. However, compared to other kinds of methods, it is generally underperforming. One common observation is the reconstruction network can also reconstruct the anomalies. For instance, Masked autoencoders can reconstruct the input with a masking ratio of more than 75% on the input space (surely the work is proposed for self-supervised learning, not anomaly detection). The authors' method showed reasonable performance, through feature-level masking, adding noise, and additive refinement.

However, the achieved performance is not very strong. First, it does not provide consistent improvement over different cases. On average, it is on a par with some existing methods. It can be important and interesting to discuss why the method is effective in some cases and less effective in others.

Second, the authors argued prior work is more sensitive to hyperparameter settings. It is not a strong argument in my opinion. All these methods are thresholding-based methods, including the proposed one. Thus, they all somewhat need some validation data to set the threshold before deployment, e.g., finding the value based on the expected false positive or true positive ratio. AUROC and AUPR are threshold-free evaluation metrics, but a threshold is still needed to really make a decision. In practice, it is probably not difficult to obtain some anomalies for validation. The important question is if the choice of hyperparameters is good for unseen anomalies at test time.

Third, at the training time, the model does not exploit any prior knowledge about anomaly patterns to train the reconstruction. In other words, the model decides on its own which cue to use for learning the best reconstruction strategy, largely depending on the given dataset. In other words, it is also hard to predict the model reacts to which type of anomalies. The evaluated benchmarks mainly considered either semantic or non-semantic anomalies. What if construct both semantic and non-semantic anomalies for the same in-distribution dataset, i.e., CIFAR10-dog as the inliers, CIFAR10-others as outliers, CIFAR10-corruption or some other non-semantic shift as another type of outliers.

---

> ### Author Response · Authors · 2022-08-02
> **Rebuttal for [R2] 1/2**
>
> # 1. Performance in terms of AUPRC
> Common rebuttal for [R1, R3]: Comparison with other mask-based restoration methods
>
> # 2. Is the hyperparameter sensitivity problem really important and is it resolved?
> Common rebuttal for [R2, R3]: Is the hyperparameter sensitivity problem really important and is it resolved?
>
> # 3. There are many studies showing better performance on the MVTec benchmark
> Most of the papers mentioned in "https://paperswithcode.com/sota/anomaly-detection-on-mvtec-ad" are side information based methods such as using prior knowledge(using augmented data as negative samples or assuming that anomaly regions are locally distributed, etc.) or using a pretrained model. MMRR is an unsupervised method and does not use prior knowledge or a pretrained network, so it was compared with methods that did not use these side information equally.
>
> # 4. How to handle open-set recognition setting
> The setting of multiple classes as inliers is a setting used in the research field of open-set recognition.
> And one of the key issues of open-set recognition is how to use class label information well.
> Our method is an anomaly detection study in the setting where class label information is not available. Therefore, it does not contain considerations on how to use class label information.
> We used the framework of MMRR as it is for open-set recognition experiments. And only normal data was changed to include data of multiple classes. And as expected, low performance was achieved because class information was not used. All experiments for openset recognition were taken from the settings in [1](Table 9).
> |   | MNIST(AUPRC) | MNIST(AUROC) | CIFAR10(AUPRC) | CIFAR10(AUROC) |
> |------|--------------|--------------|----------------|----------------|
> | 0    | 0.8191       | 0.7192       | 0.6537         | 0.5527         |
> | 1    | 0.8198       | 0.7448       | 0.6293         | 0.5171         |
> | 2    | 0.6707       | 0.672        | 0.6342         | 0.5502         |
> | 3    | 0.6572       | 0.6327       | 0.6682         | 0.5696         |
> | 4    | 0.8302       | 0.7491       | 0.608          | 0.4993         |
> | MEAN | 0.7594       | 0.70356      | 0.63868        | 0.53778        |
>
> # 5. OOD setting
> Comparing Cifar10 and Cifar10-corruption can be seen as a kind of OOD (out-of-distribution) setting.
> For the OOD experiment, we measured how well a network trained on the cifar10 dataset of a specific class detects the cifar10-corrupted dataset of the same class. Since the dataset size of Cifar10-corruption is too large, 1000 samples of all corruption data of the corresponding class were randomly sampled and used for the experiment.
> |     | AUROC   |
> |------|---------|
> | 0    | 0.5966  |
> | 1    | 0.5977  |
> | 2    | 0.5804  |
> | 3    | 0.5555  |
> | 4    | 0.5553  |
> | 5    | 0.5504  |
> | 6    | 0.5614  |
> | 7    | 0.6003  |
> | 8    | 0.5402  |
> | 9    | 0.6175  |
> | MEAN | 0.57553 |
>
> # 6. Adding noise vs just putting zeros
> There is a reason for using uniform sampled noise in the masking process.
> If noise is not used(just putting zeros), even if the masking level is very small, it can be very easy to recover from limited information.
> For example, even if the masking level is 0.01, if all the pixel values ​​of the mask are 0.01, the denoising autoencoder can restore the original data by just multiplying the input by 100x.
> This trivial solution occurs because a continuous mask is used to make the mask generation process differentiable.
>
> # 7. why the learned mask does not select a shortcut e.g, masking out every n pixels(like a downsampling) to achieve good reconstruction
> Since both the embedding network and the masking network utilize a U-Net architecture with skip connection, we think that features and masks of a specific dimension are highly related to data of the corresponding dimension.
> And if you look at the generated mask, you can see that the pixel value of the position where there is a salient part such as an edge is larger than the value of the pixel of the other part.
> Of course, one of the easiest masks that can restore data overall well will be a grid pattern mask that discloses information every n pixels.
> However, the grid pattern mask has the disadvantage of not concentrating the information on the salient part like edges.
> And our masking network is shown to generate a data-dependent mask because generating a mask focusing on salient parts such as edges can make the restoration loss smaller than grid pattern mask.
>
> ## Reference
> [1] Moon, WonJun, et al. "Difficulty-Aware Simulator for Open Set Recognition." arXiv preprint arXiv:2207.10024 (2022).

---

> ### Author Response · Authors · 2022-08-02
> **Rebuttal for [R2] 2/2**
>
> # 8. In Fig 3, there is no reconstruction difference between normal data and abnormal data.
> Referring to Fig. 3, it can be seen that the deer (abnormal data) restored through MMRR has a completely different color from the original deer. And it actually shows a high refinement loss compared to the ship(normal) data.
> In this example, it can be seen that MMRR performed anomaly detection by focusing on color rather than shape.
> This is a natural phenomenon because we did not force MMRR to perform anomaly detection through semantic elements by using prior knowledge or pretrained models.
> Therefore, the MMRR decides which criteria to use to perform anomaly detection on its own based on the data.(agree with “the model decides on its own which cue to use for learning the best reconstruction strategy, largely depending on the given dataset” mentioned by [R2]).
> As such, MMRR can perform anomaly detection in a direction different from the human perspective.
> But on the contrary to the previous example,  in the case of simple aligned data such as MNIST in Fig. 3, anomaly detection is performed in a direction similar to human perspective.
> More qualitative results can be found in the supplementary material.
>
> # 9. it is still unclear on which conditions/reasons the model is particularly good and in which scenarios it is expected to be underperforming.
> The intrinsic complexity mentioned in describing the refinement module will be the key to this question.
> In the case of digit 1 of MNIST, it can be seen that restoration through MMRR is well performed even though it is abnormal data.
> As mentioned in the MNIST experiment section, the reason for this phenomenon was thought to be that the intrinsic complexity of digit 1 was simple compared to other digits.
> The low intrinsic complexity means that the amount of information required for restoration is small, which leads to the result of good restoration even at a small masking level.
> Here, suppose that digit 8, which is expected to have high intrinsic complexity, is normal data and digit 1 is abnormal data.
> Although digit 8 is normal data, the network requires high masking level (a lot of information) to restore 8. On the other hand, even though digit 1 is abnormal data, it will be well restored even at a low masking level (small amount of information).
> In the worst case, if the intrinsic complexity of digit 1 is simple enough that the refinement module cannot cover it, and the restoration is good even at a very low masking level, MMRR fails to detect anomaly.
> In summary, although the refinement module resolves the intrinsic complexity problem to some extent, MMRR is not completely free from the problem of being easy to reconstruct data with low intrinsic complexity.
> Similarly, in the CIFAR10 dataset, classes such as ships and airplanes consist of data that is relatively intrinsically simpler to restore when compared to data of other classes. This is because the data of the ship or airplane class consists of small foreground objects with a monotonous background.
> Now we can easily infer that when airplane or ship with low intrinsic complexity is normal data, anomaly detection performance will be higher than when other classes are normal.

---

> > ### Comment · Reviewer_rzLV · 2022-08-08
> > **response to rebuttal**
> >
> > Thank you for the answers. I have decided to keep my original rating.
> >
> > I am still not convinced by how competitive this approach is. More importantly, considering outliers can be in countless forms, it is fine to exploit some meta-level knowledge for designing anomaly detection algorithms, e.g., aiming at semantic or non-semantic outliers. While the authors argued their approach is complete unsupervised, their model is biased towards some form of anomalies (not all forms), and the more crucial problem is they have no control and awareness of which from the model learnt. Therefore, in my opinion, it is actually weaker than approaches that explicitly exploit the meta concept of anomalies and make sure their methods react to that type of anomalies. Here, the meta concept does not require collecting anomaly data for training, it could be as high-level as to detect any outliers do not belong the same semantic classes (e.g., one vs. rest in CIFARs), or outliers with abnormal local patterns or structures (industry defect).

---

> > > ### Author Response · Authors · 2022-08-09
> > > **response to [R2]**
> > >
> > > Before answering your comment, thank you very much for R2's response.
> > >
> > > When meta-level knowledge is used, there is a clear advantage in that it is easy to interpret because anomaly detection is performed in terms of the used meta-knowledge. However, as we have mentioned, this meta-level knowledge is not generally applicable. For example, the meta-level knowledge of geometric transform introduced in [1] is a very good approach in the situation where the foreground object detects anomalies that are significantly different in terms of shape. However, this method experimentally found and used only specific transformations such as rotation, translation, and flip, which are helpful in the performance of anomaly detection. And it was mentioned in the paper that there is a performance degradation when applying the affine transform that can be used in general. (Can it be said that the process of finding out which transformation should be used to perform anomaly detection well does not use information about anomaly data?) And since [1] is applicable only to images, [2] applied the affine transform to apply the transform-based method even on tabular datasets (thyroid, arrythmia). However, it can be seen that the performance is significantly lower than our performance seen in "Rebuttal for R3 5". Also, as mentioned above(Common rebuttal for [R1, R3]: Comparison with other mask-based restoration methods
> > > ), the method that assumes a local defect area as in [3] shows poor performance in anomaly detection in CIFAR10.
> > > |     | thyroid(F1) | arrhythmia(F1) |
> > > |------|-------------|----------------|
> > > | [2]  | 0.745       | 0.52           |
> > > | Ours | 0.8596      | 0.7427         |
> > >
> > > Papers such as [4] and [5], which show the best performance among our comparison targets, do not compare methods using meta-knowledge. Papers such as [4] and [5], which showed the best performance among our comparison targets, did not compare their methods with those using meta-knowledge.
> > > It is true that anomaly detection methods using meta-knowledge are easy to interpret and show good performance. However, methods that try to solve anomaly detection without using prior knowledge about data must be respected as a solution with a different approach(basic approach).
> > >
> > > Additionally, R2 noted that our method was biased because the performance in cifar10 was class-dependent. However, as we mentioned in (Rebuttal for R2 2/2 9), it is a performance problem caused by the intrinsic complexity difference of each class of cifar10, not because our method is biased.
> > >
> > > [1] Golan, Izhak, and Ran El-Yaniv. "Deep anomaly detection using geometric transformations." Advances in neural information processing systems 31 (2018). \
> > > [2] Bergman, Liron, and Yedid Hoshen. "Classification-based anomaly detection for general data." arXiv preprint arXiv:2005.02359 (2020). \
> > > [3] Li, Chun-Liang, et al. "Cutpaste: Self-supervised learning for anomaly detection and localization." Proceedings of the IEEE/CVF Conference on Computer Vision and Pattern Recognition. 2021. \
> > > [4] Goyal, Sachin, et al. "DROCC: Deep robust one-class classification." International Conference on Machine Learning. PMLR, 2020.\
> > > [5] Venkataramanan, Shashanka, et al. "Attention guided anomaly localization in images." European Conference on Computer Vision. Springer, Cham, 2020.

---

### Official Review · Reviewer_UWD3 · 2022-07-11

**Rating:** 2
**Confidence:** 5
**Soundness:** 1 poor
**Presentation:** 2 fair
**Contribution:** 1 poor

**Summary:**

This paper presents an unsupervised learning based anomaly detection method, which masks the images at multi-levels for restoration and then refine the restored images with refinement network. To overcome hyperparameter sensitivity, this work ensembles multiple restoration results from different level of masks and propose a novel mask generation and refinement method to achieve hyperparameter robustness.  Experiments are conducted on  several  benchmarks

**Questions:**

What is the advantage of the proposed mask design method in comparison with the masking design in other works? Although the hypermeter issue is relieved to some degree, the restoration quality of designed mask scheme is worse than the peer works.



**Limitations:**

The paper didn't address the limitation and potential negative societal impact of the work.


**Strengths And Weaknesses:**

Strengths
1.This work proposed a multi-level masking scheme for image restoration and ensemble restoration results to overcome hyperparameter sensitivity.
2.The experiments are evaluated on multiple datasets.

Weaknesses
1.The novelty of this work is very limited. The mask-based restoration has been studied by search works, and they employed the ensembles of restoration results of different scale masks or different pattern masks.
2.Some important related works are missing in the paper, and this work does not compare the AD performance with these works. E.g. “CFLOW-AD: Real-Time Unsupervised Anomaly Detection with Localization via Conditional Normalizing Flows”, “Learning Semantic Context from Normal Samples for Unsupervised Anomaly Detection” etc.
3.The proposed method adopts a complicated framework including multiple restoration and refinement networks, but the AD performance on MVTEC and Cifar-10 dataset is far below the state-of-the-art.

---

> ### Author Response · Authors · 2022-08-01
> **Rebuttal for [R1]**
>
> # 1. Comparison with other mask-based restoration methods
> Common rebuttal for [R1, R3]: Comparison with other mask-based restoration methods
>
> # 2. AD performance with [1]
> [1] belongs to the category of the method using the pre-trained model among the side-information-based methods of the related work we introduced. The unsupervised anomaly detection methods we defined are methods that do not use prior knowledge or a pre-trained model trained from external data. We compared the MMRR with methods belonging to the unsupervised anomaly detection method we defined. This is because prior knowledge or pretrained models provide a very powerful hint (representation) in anomaly detection.
>
> # 3. Restoration quality of designed mask scheme is worse than other methods
> In anomaly detection, the difference in restoration loss between normal and abnormal data is more important than absolute restoration quality. In order for anomaly detection to be successful, the restoration loss (refinement loss in our case) needs to be small compared to the abnormal data without requiring that the restoration loss of the normal data is absolutely small. In addition to Fig 3., if you look at the supplementary, you can see more examples of restoration at various masking levels. And through this, it can be seen that the abnormal data is not restored well compared to the normal data.
>
> # 4. The paper didn't address the limitation
> Our method has two limitations. First, as seen in the MNIST experiment section, it is difficult to detect anomaly in the case of abnormal data that is already sufficiently restored even with a low masking level. This is a difficult problem to solve even if a refinement module is used to overcome the inherent complexity difference.
> Second, as shown in the conclusion section, MMRR has the disadvantage of large computation because it ensembles the degree of reconstruction at various masking levels in the evaluation process.
>
> ## Reference
> [1].  ”CFLOW-AD: Real-Time Unsupervised Anomaly Detection with Localization via Conditional Normalizing Flows” WACV2022

---

> ### Author Response · Authors · 2022-08-10
> **The discussion ended**
>
>
> Author and reviewer discussion ended. Could you please go over our rebuttal and check the responses? We believe that we have addressed all your concerns and that including these discussions will further strengthen our paper. We hope you reflect this in your final review and the score. We thank you again for your time and efforts in reviewing our paper.

---

### Author Response · Authors · 2022-08-01
**Common rebuttals**

We first designate the reviewer numbers as follows.

R1: Reviewer UWD3\
R2: Reviewer rzLV\
R3: Reviewer 7z6Y

# Common rebuttal for [R1, R3]: Comparison with other mask-based restoration methods
## 1. Comparison with [1]
The large framework of erasing a part of data and performing restoration using a learnable mask can be said to be similar. However, the implicit intuition and detailed method are completely different. First, in the case of [1], the most difficult part to restore is deleted from the data, and then that part is restored. And in the case of [1], in order to generate a mask that erases important parts, adversarial training is conducted between the mask generating network that wants to increase the restoration loss and the restoration network that wants to reduce the restoration loss. Conversely, in the case of MMRR, an essential part is left for restoration and the rest is restored. In addition, MMRR is robust to hyperparameters by eliminating the element of adversarial training through a new mask generation method and ensembles the degree of restoration at various limitation levels. The better performance than [1] of MMRR shows that the mask method of MMRR is efficient.
## 2. Comparison with [2]
[2] tried to better learn the semantic features of the data through the process of reconstructing the masked data with predefined masks with different grid sizes. And using the trained inpainting network, anomaly score map was generated through an iterative process called mask refinement in the inference process.
The biggest difference between this study and MMRR is whether the mask is learnable. In case of [2], a non-learnable predefined mask is used, whereas the mask of MMRR is learned to erase the remaining parts while leaving the parts essential for restoration at the corresponding masking level.
[2] has good performance in MVTecAD. However, this study using predefined masks similar to checkboard seems to assume that the abnormal part is locally distributed. This assumption may work well for the MVTecAD dataset. However, this assumption may not be valid for data such as CIFAR10. This is because, in the case of anomaly detection in CIFAR10, data of different classes must be detected. And in this case, the abnormal element in data may not be locally distributed. For example, CutPaste[26], one of the studies that also assumed a local abnormal situation, showed good performance in MVTecAD. But showed a relatively low performance of 69.4 AUC in the CIFAR10 dataset. On the other hand, MMRR makes no assumptions about abnormal elements in designing the mask, and mask is learnable. This means that an appropriate mask is created according to the characteristics of the data.
## 3. Comparison with [3]
[3] Also, like [2], it has a big difference from MMRR in that it uses a predefined mask. And it can be seen that our learnable mask is efficient because MMRR is better than [3] in terms of performance on CIFAR10 and MVTecAD datasets.


## Reference
[1]. “One-Class Learned Encoder-Decoder Network with Adversarial Context Masking for Novelty Detection” WACV 2022\
[2]. “Self-Supervised Masking for Unsupervised Anomaly Detection and Localization” IEEE Transactions on Multimedia 2022\
[3]. “Learning Semantic Context from Normal Samples for Unsupervised Anomaly Detection” AAAI 2021

---

### Author Response · Authors · 2022-08-02
**Common rebuttals**

We first designate the reviewer numbers as follows.

R1: Reviewer UWD3\
R2: Reviewer rzLV\
R3: Reviewer 7z6Y

# Common rebuttal for [R2, R3]: Is the hyperparameter sensitivity problem really important and is it resolved?
We agree with the [R2] that the threshold should be set in order to actually use the proposed AD method as a binary classifier. We also agree that AUROC and AUPRC are threshold-free evaluation metrics and have limitations as indicators that show whether the AD method can actually do binary classification well. However, we assume strict unsupervised anomaly detection setting in which no anomaly data can be used in the training process (including the validation process). Unsupervised anomaly detection is a basic study for performing anomaly detection. And it is important whether an algorithm learned using only normal data has the inherent ability to distinguish between normal data and abnormal data well.

Most of the existing unsupervised anomaly detection methods(no prior knowledge, no pretrained models) try to limit the capacity of the generative model or try to find a hypersphere that covers only normal data. The most important point we would like to argue is that existing methods are highly dependent on the hyperparameters that are inevitably generated to achieve the above goals. As mentioned in the hyperparameter sensitivity section, although [1] is a SOTA method, it can be seen that the AUROC performance varies greatly depending on the radius value (see Fig 8. in [1]). Here, the radius value is a hyperparameter that greatly affects the learning direction of the network that determines the range of normal data in the feature space. And this phenomenon occurs not only in DROCC but also in most other recent studies(For example, Fig 9. in [2], Table 4. in [3]). And, of course, the reported performance of existing studies is the performance with hyperparameters where their method shows the best anomaly detection performance. And it can be said that the existing studies performed tuning using abnormal data to find the hyperparameter that their method best performs anomaly detection.

However, in the case of MMRR, even though it has conflicting goals of information restriction and restoration(for designing generative models with limited capacity), the novel mask generation method allows all learning processes to be performed using only one objective function (step1: restoration loss, step2: refinement loss). This means that there are no hyperparameters that can significantly affect the learning results of the network in the case of MMRR compared to previous studies.

Of course, we agree with [R3] that MMRR cannot be said to be robust in terms of hyperparameters used in model design or hyperparameters of optimizer. However, such hyperparameters cannot be considered to have a large effect when compared with the hyperparameters that balances the loss functions that directly affects the training results of model.This can be confirmed through the fact that the experimental results using a wide-residual network based UNet(WideUNet) and the normal UNet are not significantly different.

|        | WideUNet | UNet  |
|---------|----------|-------|
| MNIST   | 0.967    | 0.972 |
| FMNIST  | 0.93     | 0.931 |
| CIFAR10 | 0.737    | 0.735 |

In summary, our method does not make any effort to find hyperparameters that have the best anomaly detection performance in the evaluation process with the help of abnormal data. Therefore, it can be said that it is a meaningful result that MMRR showed comparable performance to existing studies that found and applied a hyperparameter with the best anomaly detection performance with the help of abnormal data.

## Reference
[1]. Goyal, Sachin, et al. "DROCC: Deep robust one-class classification." International Conference on Machine Learning. PMLR, 2020.\
[2]. Akçay, Samet, Amir Atapour-Abarghouei, and Toby P. Breckon. "Skip-ganomaly: Skip connected and adversarially trained encoder-decoder anomaly detection." 2019 International Joint Conference on Neural Networks (IJCNN). IEEE, 2019.\
[3]. J. Hou, Y. Zhang, Q. Zhong, D. Xie, S. Pu, and H. Zhou. Divide-and-assemble: Learning block-wise memory for unsupervised anomaly detection. In Proceedings of the IEEE/CVF International Conference on Computer Vision, pages 8791–8800, 2021.

---

### Author Response · Authors · 2022-08-02
**Common rebuttals**

We first designate the reviewer numbers as follows.

R1: Reviewer UWD3\
R2: Reviewer rzLV\
R3: Reviewer 7z6Y

# Common rebuttal for [R2, R3]: Performance in terms of AUPRC
Performance in terms of AUPRC and AUROC can be seen in the following table.
|      | MNIST   | MNIST   | FMNIST  | FMNIST  | CIFAR10 | CIFAR10 |
|------|---------|---------|---------|---------|---------|---------|
|      | AUROC   | AUPRC   | AUROC   | AUPRC   | AUROC   | AUPRC   |
| 0    | 0.9941  | 0.9446  | 0.9105  | 0.66    | 0.7965  | 0.412   |
| 1    | 0.9982  | 0.9748  | 0.9917  | 0.9666  | 0.7377  | 0.3148  |
| 2    | 0.94    | 0.6045  | 0.8831  | 0.5005  | 0.7024  | 0.2282  |
| 3    | 0.955   | 0.8755  | 0.94    | 0.7263  | 0.6595  | 0.1818  |
| 4    | 0.9352  | 0.6214  | 0.8993  | 0.5305  | 0.7817  | 0.3031  |
| 5    | 0.971   | 0.8801  | 0.948   | 0.724   | 0.6739  | 0.2059  |
| 6    | 0.989   | 0.8254  | 0.8031  | 0.324   | 0.7641  | 0.2775  |
| 7    | 0.966   | 0.7932  | 0.9914  | 0.9171  | 0.7037  | 0.2345  |
| 8    | 0.945   | 0.6254  | 0.9206  | 0.7193  | 0.8181  | 0.4065  |
| 9    | 0.98    | 0.9054  | 0.9837  | 0.8423  | 0.7325  | 0.2874  |
| MEAN | 0.96735 | 0.80503 | 0.92714 | 0.69106 | 0.73701 | 0.28517 |

---

### Meta-Review · Area_Chair_nC8e · 2022-08-23

**Recommendation:** Reject
**Confidence:** Certain

**Metareview:**

The paper proposes a Multi-Level Masking and Restoration with Refinement  to solve the hyperparameter sensitivity problem in anomaly detection studies. Reviewers had some concerns regarding this work including limited novelty, inconsistent numerical evaluation with prior works, lack of  discussions on benefits of using prior knowledge, etc. I appreciate that the authors were active during the rebuttal period to address these concerns but I think the paper needs a bit more work before being accepted.

**Award:**

No

---

### Decision · Program_Chairs · 2022-09-14

Reject